# Fast Rates in Stochastic Online Convex Optimization by Exploiting the Curvature of Feasible Sets

**Taira Tsuchiya**
The University of Tokyo and RIKEN
tsuchiya@mist.i.u-tokyo.ac.jp

**Shinji Ito**
The University of Tokyo and RIKEN
shinji@mist.i.u-tokyo.ac.jp

## Abstract

In this work, we explore online convex optimization (OCO) and introduce a new condition and analysis that provides fast rates by exploiting the curvature of feasible sets. In online linear optimization, it is known that if the average gradient of loss functions exceeds a certain threshold, the curvature of feasible sets can be exploited by the follow-the-leader (FTL) algorithm to achieve a logarithmic regret. This study reveals that algorithms adaptive to the curvature of loss functions can also leverage the curvature of feasible sets. In particular, we first prove that if an optimal decision is on the boundary of a feasible set and the gradient of an underlying loss function is non-zero, then the algorithm achieves a regret bound of $O(\rho \ln T)$ in stochastic environments. Here, $\rho > 0$ is the radius of the smallest sphere that includes the optimal decision and encloses the feasible set. Our approach, unlike existing ones, can work directly with convex loss functions, exploiting the curvature of loss functions simultaneously, and can achieve the logarithmic regret only with a local property of feasible sets. Additionally, the algorithm achieves an $O(\sqrt{T})$ regret even in adversarial environments, in which FTL suffers an $\Omega(T)$ regret, and achieves an $O(\rho \ln T + \sqrt{C \rho \ln T})$ regret in corrupted stochastic environments with corruption level $C$. Furthermore, by extending our analysis, we establish a matching regret upper bound of $O\left(T^{\frac{q-2}{2(q-1)}} (\ln T)^{\frac{q}{2(q-1)}}\right)$ for $q$-uniformly convex feasible sets, where uniformly convex sets include strongly convex sets and $\ell_p$-balls for $p \in [2, \infty)$. This bound bridges the gap between the $O(\ln T)$ bound for strongly convex sets ($q = 2$) and the $O(\sqrt{T})$ bound for non-curved sets ($q \to \infty$).

## 1 Introduction

This paper considers online convex optimization (OCO), a framework in which a learner and an environment interact in a sequential manner. At the beginning, a convex body (or feasible set) $K \subseteq \mathbb{R}^d$ is given. At each round $t = 1, \dots, T$, the learner selects a decision $x_t \in K$ from the convex body $K$ using information obtained up to round $t - 1$. Then, the environment determines a convex loss function $f_t \colon K \to \mathbb{R}$, and the learner suffers loss $f_t(x_t)$ and observes $\nabla f_t(x_t) \in \mathbb{R}^d$. The goal of the learner is to minimize the regret, which is the expectation of the difference between the cumulative loss of decisions $(x_t)_{t=1}^T$ and that of a single optimal decision $x_\star$ fixed in hindsight, that is, $\mathsf{R}_T = \mathbb{E}\left[\sum_{t=1}^T (f_t(x_t) - f_t(x_\star))\right]$ for $x_\star = \arg\min_{x \in K} \mathbb{E}\left[\sum_{t=1}^T f_t(x)\right]$. OCO is called online linear optimization (OLO) when $(f_t)_t$ are linear functions, *i.e.*, $f_t(\cdot) = \langle g_t, \cdot \rangle$ for some $g_t \in \mathbb{R}^d$.

In OCO and OLO, the well-known online gradient descent (OGD) achieves an $O(\sqrt{T})$ regret upper bound for Lipschitz continuous $f_t$ [27]. In general, this upper bound cannot be improved and is known to match the $\Omega(\sqrt{T})$ regret lower bound [8]. However, this lower bound can be circumvented under certain conditions. The most typical way is to exploit the curvature of loss functions. It is known

38th Conference on Neural Information Processing Systems (NeurIPS 2024).

**Table 1:** Comparison of our regret upper bounds with existing bounds. All bounds assume that loss functions are $G$-Lipschitz (except Lines 1–3) and $x_\star$ is on the boundary of $K$. The upper bounds that contain the variable $L > 0$ assume $\|g_1 + \cdots + g_t\|_2 \geq tL$ for all $t \in [T]$. We use $f^\circ = \mathbb{E}_{f \sim \mathcal{D}}[f]$, $C \geq 0$ is the corruption level, and the $\tilde{\Omega}$ notation ignores logarithmic factors. The $(\kappa, 2)$-uniformly convex set is $\kappa$-strongly convex. Theorem is abbreviated as as Thm, Corollary as Cor, and sphere-enclosed as sphere-enc. Note that regret bounds proven in this study can be simultaneously achieved by the same algorithm with identical parameters.

| Reference | Feasible set | Loss functions | Regret bound |
|---|---|---|---|
| [9], **This work** (Thm 8) | ellipsoid $W_\lambda$ in Section 2.3.2 ($\lambda$-strongly convex) | $f_t(\cdot) = \langle h_t^L, \cdot \rangle$ in Thm 6 | $\Omega\left(\dfrac{1}{\lambda L} \ln T\right)$, $\Omega\left(\dfrac{1}{\lambda \|\nabla f^\circ(x_\star)\|_2} \ln T\right)$ |
| **This work** (Thm 9) | | corrupted | $\tilde{\Omega}\left(\dfrac{1}{\lambda \|\nabla f^\circ(x_\star)\|_2} + \sqrt{\dfrac{C}{\lambda \|\nabla f^\circ(x_\star)\|_2}}\right)$ |
| **This work** (Cor 12) | | $f_t(\cdot) = \langle h_t^L, \cdot \rangle$ in Thm 6 | $O\left(\dfrac{1}{\lambda L} \ln T\right)$ |
| **This work** (Thms 10, 14) | $(\rho, x_\star, f^\circ)$-sphere-enc. | stochastic, convex | $O\left(\dfrac{G^2 \rho}{\|\nabla f^\circ(x_\star)\|_2} \ln T\right)$ |
| **This work** (Thm 13) | $(\rho, \widetilde{x}_\star, \widetilde{f}^\circ)$-sphere-enc. | corrupted, convex | $O\left(\dfrac{G^2 \rho}{\|\nabla \widetilde{f}^\circ(\widetilde{x}_\star)\|_2} \ln T + \sqrt{\dfrac{C G^2 \rho}{\|\nabla \widetilde{f}^\circ(\widetilde{x}_\star)\|_2} \ln T}\right)$ |
| Huang et al. [9] | $\lambda$-strongly convex | adversarial, linear | $O\left(\dfrac{G^2}{\lambda L} \ln T\right)$ |
| Molinaro [17] | | adversarial, linear | $O\left(\dfrac{G c'}{\lambda} \ln T\right)$ $\left(c' = \max_{x \in \mathbb{R}^d_{>0} : \|x\|=1} \langle u, x \rangle\right)$ |
| **This work** (Thm 15) | | stochastic, convex | $O\left(\dfrac{G^2}{\lambda \|\nabla f^\circ(x_\star)\|_\star} \ln T\right)$ |
| Kerdreux et al. [11] | $(\kappa, q)$-uniformly convex | adversarial, linear | $O\left(\dfrac{G^{\frac{q}{q-1}}}{(\kappa L)^{\frac{1}{q-1}}} T^{\frac{q-2}{q-1}}\right)$ |
| **This work** (Thm 15) | | stochastic, convex | $O\left(\dfrac{G^{\frac{q}{q-1}}}{(\kappa \|\nabla f^\circ(x_\star)\|_\star)^{\frac{1}{q-1}}} T^{\frac{q-2}{2(q-1)}} (\ln T)^{\frac{q}{2(q-1)}}\right)$ |

that OGD with a learning rate of $\Theta(1/t)$ and online Newton step (ONS) can achieve an $O(\frac{1}{\alpha} \ln T)$ and $O(\frac{d}{\beta} \ln T)$ regret for $\alpha$-strongly-convex and $\beta$-exp-concave loss functions, respectively [8].

Another way to circumvent the lower bound is to harness *the curvature of the feasible set $K$*. Existing studies proved that in OLO if the feasible set is curved and loss vectors $g_t$ are biased towards a specific direction, the follow-the-leader (FTL) algorithm can achieve a logarithmic regret. In particular, Huang et al. [9] first proved that under the *growth condition* that there exists $L > 0$ such that $\|g_1 + \cdots + g_t\|_2 \geq tL$ for any $t \in [T] = \{1, \ldots, T\}$, FTL achieves an $O(\frac{G^2}{\lambda L} \ln T)$ regret for $\lambda$-strongly convex $K$ and $G$-Lipschitz loss functions. This bound matches their lower bound of $\Omega(\frac{1}{\lambda L} \ln T)$. Molinaro [17] also proves that FTL can achieve a logarithmic regret under the different assumption on the loss vectors that $g_t \leq 0$ for all $t \in [T]$, providing an intuitive and simple proof.

Their approach, however, has several remaining limitations. First, they only consider OLO. While the linearization technique allows us to solve OCO by OLO, this may prevent us from leveraging the curvature of loss functions. Second, their analysis requires the curvature over the entire boundary of the feasible set, which is a rather limited condition. Finally, some of their approach suffers an $\Omega(T)$ regret if the ideal conditions on loss vectors, such as the growth condition, are not satisfied. Note that we cannot know in advance whether such conditions are satisfied or not. Exceptions are the method based on the expert tracking algorithm in [9, Section 4], in which FTL is combined with follow-the-regularized-leader, and the work by Anderson and Leith [1], who investigated the online lazy gradient descent over the strongly convex sets.

To overcome these limitations, we consider using algorithms adaptive to the curvature of loss functions [21, 22, 24], also known as *universal online learning*. The original motivation of this line of

work is to automatically achieve a regret bound that depends on the true curvature level of loss functions, *e.g.,* parameters of strong convexity or exp-concavity, without knowing them. The crux of their analysis is to derive a bound of $\sum_{t=1}^{T}\langle\nabla f_t(x_t), x_t - x_\star\rangle = O\big(\sqrt{\sum_{t=1}^{T}\|x_t - x_\star\|_2^2 \ln T}\big)$.

**Contributions of this paper**  We introduce a new condition for achieving fast rates in OCO. We first show that algorithms adaptive to the curvature of loss functions can exploit the curvature of feasible sets and overcome the three limitations mentioned earlier. We prove the following theorem:

**Theorem 1** (informal version of Theorems 10 and 13)**.** *Any algorithm with $\sum_{t=1}^{T}\langle\nabla f_t(x_t), x_t - x_\star\rangle = O\big(c_{\mathsf{sc}}\sqrt{\sum_{t=1}^{T}\|x_t - x_\star\|_2^2 \ln T}\big)$ for some $c_{\mathsf{sc}} > 0$ achieves $\mathsf{R}_T = O\Big(\frac{c_{\mathsf{sc}}^2 \rho}{\|\nabla f^\circ(x_\star)\|_2} \ln T\Big)$ in stochastic environments, where $f^\circ = \mathbb{E}_{f_t}[f_t]$ and $\rho > 0$ is the smallest radius of a sphere that includes $x_\star$ and encloses $K$. The same algorithm achieves $\mathsf{R}_T = O(\rho \ln T + \sqrt{C\rho \ln T})$ in corrupted stochastic environments for corruption level $C$ and $\mathsf{R}_T = O(\sqrt{T})$ in adversarial environments.*

This upper bound matches an existing lower bound [9, Theorem 9], specifically when considering the environment employed to construct their lower bound. This will be formally stated in Corollary 12.

The advantage of our approach over the existing approach is that it overcomes all three limitations of the existing approach mentioned earlier. That is, (i) in contrast to existing studies, it can work with OCO without the linearization, allowing us to simultaneously exploit the curvature of feasible sets and the curvature of loss functions (see Theorem 14). (ii) Even in worst cases, where the specific conditions on loss vectors, such as the growth condition, are not satisfied, an $O(\sqrt{T})$ regret upper bound can be achieved. (iii) The local structure of $K$ around optimal decision $x_\star$ is sufficient for our approach to achieve the logarithmic regret. As a further advantage, our approach can achieve an $O(\rho \ln T + \sqrt{C\rho \ln T})$ regret bound for corrupted stochastic environments with corruption level $C \geq 0$, which are intermediate environments between stochastic and adversarial environments (Theorem 13). We provide a regret lower bound that nearly matches this upper bound (Theorem 9).

Our approach can also be used to obtain fast rates on *uniformly convex* feasible sets, a broader class that includes *strongly convex* sets and $\ell_p$-balls for $p \in [2, \infty)$. For $q$-uniformly convex $K$, Kerdreux et al. [11] proves an regret bound of $O\big(T^{\frac{q-2}{q-1}}\big)$, which is smaller than $O(\sqrt{T})$ only when $q \in (2, 3)$. We improve this bound by proving the following upper bound, which matches the lower bound in [2]:

**Theorem 2** (informal version of Theorem 15)**.** *In online convex optimization with $q$-uniformly convex feasible set $K$, the same algorithm as Theorem 1 achieves $\mathsf{R}_T = O\big(T^{\frac{q-2}{2(q-1)}}(\ln T)^{\frac{q}{2(q-1)}}\big)$.*

This becomes a fast rate for any $q > 2$ and is strictly better than the bound in [11]. Our bound interpolates between the $O(\ln T)$ bound for strongly convex sets (when $q = 2$) and the $O(\sqrt{T})$ bound for non-curved feasible sets (when $q \to \infty$). Table 1 summarizes the regret comparison.

## 2   Preliminaries

Let $e_i \in \{0, 1\}^d$ be the $i$-th standard basis of $\mathbb{R}^d$, and $\mathbf{1}$ be the all-one vector. For $p \in [1, \infty]$ and vector $x$, let $\|x\|_p$ be $\ell_p$-norm. Let $\xi > 0$ be a constant satisfying $\|x\|_2 \leq \xi\|x\|$ for any $x \in \mathbb{R}^d$. For a norm $\|\cdot\|$, we use $\|x\|_\star = \sup\{\langle x, y\rangle \colon \|y\| \leq 1\}$ to denote its dual norm. Let $\mathbb{B}_{\|\cdot\|}(x, r)$ be a ball with radius $r$ centered at $x$ associated with $\|\cdot\|$, *i.e.,* $\mathbb{B}_{\|\cdot\|}(x, r) = \{z \colon \|z - x\| \leq r\}$. We use $\mathbb{B}(x, r)$ to denote the Euclidean ball with radius $r$ centered at $x$ and $\mathbb{B}_{\|\cdot\|}$ to denote the unit ball. Let $\mathrm{bd}(K)$ be the boundary of $K$. A function $f \colon \mathbb{R}^d \to (-\infty, \infty]$ is convex if for all $x \in \mathrm{int\,dom}\, f$, $f(y) \geq f(x) + \langle\nabla f(x), y - x\rangle$ for all $y \in \mathbb{R}^d$.[1] For $\alpha > 0$, $f \colon K \to (-\infty, \infty]$ is $\alpha$-strongly convex over $K \subseteq \mathrm{dom}\, f$ w.r.t. $\|\cdot\|$ if for all $x, y \in K$, $f(y) \geq f(x) + \langle\nabla f(x), y - x\rangle + \frac{\alpha}{2}\|x - y\|^2$. For $\beta > 0$, $f \colon K \to (-\infty, \infty]$ is $\beta$-exp-concave if $\exp(-\beta f(x))$ is concave.

---

[1]For simplicity, this paper only considers the case that loss functions $f_t$'s are differentiable, but one can extend all the results to the subdifferentiable case in a straightforward manner.

## 2.1 Online convex optimization

We consider online convex optimization (OCO). In OCO, a convex body (or feasible set) $K \subseteq \mathbb{R}^d$ is given before the game starts. Let $D = \max_{x,y \in K} \|x - y\|_2$ be the diameter of $K$. At each round $t \in [T]$, the learner selects a decision $x_t \in K$ using information obtained up to round $t - 1$, and a convex loss function $f_t \colon K \to \mathbb{R}$ is determined by the environment. The learner then suffers a loss $f_t(x_t)$ and observes $\nabla f_t(x_t) \in \mathbb{R}^d$. The goal of the learner is to minimize the regret, which is defined as $\mathsf{R}_T = \mathbb{E}\left[\sum_{t=1}^{T} (f_t(x_t) - f_t(x_\star))\right]$ for the optimal decision $x_\star = \arg\min_{x \in K} \mathbb{E}\left[\sum_{t=1}^{T} f_t(x)\right]$. When loss functions are restricted solely to linear functions, that is, when $f_t(\cdot) = \langle g_t, \cdot \rangle$ for some $g_t \in \mathbb{R}^d$, OCO is referred to as online linear optimization (OLO).

## 2.2 Assumptions on loss functions

In this study, we assume that $f_t$ is $G$-Lipschitz, *i.e.,* $\sup_{x \in K} \|\nabla f_t(x)\|_2 \leq G$. In the following, we list three assumptions on how a sequence of $f_1, \ldots, f_T$ is generated. In stochastic environments, $f_t$ is sampled in an i.i.d. manner from a certain probability distribution $\mathcal{D}$. The expectation of $f_t$ is denoted as $f^\circ = \mathbb{E}_{f \sim \mathcal{D}}[f]$. In adversarial environments, $f_t$ is arbitrarily determined depending on the past history, and $f_t$ may depend on $x_t$. The corrupted stochastic environment is an intermediate setting between stochastic and adversarial environments. The motivation for considering this environment is that in real-world problems, a sequence of loss functions is neither stochastic nor (fully) adversarial. In this environment, at each round $t \in [T]$, $\tilde{f}_t \sim \tilde{\mathcal{D}}$ is obtained according to a certain distribution $\tilde{\mathcal{D}}$, where the expectation of $\tilde{f}_t$ is defined by $\tilde{f}^\circ = \mathbb{E}_{\tilde{f} \sim \tilde{\mathcal{D}}}[\tilde{f}]$. Then, possibly depending on $\tilde{f}_t$ and the past history, loss function $f_t$ is determined by the environment so that $\mathbb{E}\left[\sum_{t=1}^{T} \|f_t - \tilde{f}_t\|_\infty\right] \leq C$, where $\mathbb{E}\left[\sum_{t=1}^{T} \|f_t - \tilde{f}_t\|_\infty\right]$ is the corruption level. In this paper, we consider these three environments.

## 2.3 Exploiting the curvature of feasible sets

We start by introducing the definition of strongly and uniformly convex sets. We then define a new notion of convex bodies, *sphere-enclosed set*, for which we can also achieve the fast rates of $O(\ln T)$. We finally discuss the existing lower bound when exploiting the curvature.

### 2.3.1 Strong convexity and sphere-enclosedness

One common way to describe the curvature of a convex body is with the following strong convexity.

**Definition 3.** A convex body $K$ is $\lambda$-*strongly convex w.r.t. a norm* $\|\cdot\|$ if for any $x, y \in K$ and any $\theta \in [0, 1]$, it holds that $\theta x + (1 - \theta)y + \theta(1 - \theta)\frac{\lambda}{2}\|x - y\|^2 \cdot \mathbb{B}_{\|\cdot\|} \subseteq K$.

For example, $\ell_p$-balls for $p \in [1, 2]$ are $(p - 1)/2$-strongly convex w.r.t. $\|\cdot\|_p$ [7, Theorem 2], and another various examples of strongly convex sets can be found in [6, Section 5]. A more general notion of the curvature is by the following uniform convexity:

**Definition 4.** A convex body $K$ is $(\kappa, q)$-*uniformly convex w.r.t. a norm* $\|\cdot\|$ *(or q-uniformly convex)* if for any $x, y \in K$ and any $\theta \in [0, 1]$, it holds that $\theta x + (1 - \theta)y + \theta(1 - \theta)\kappa\|x - y\|^q \cdot \mathbb{B}_{\|\cdot\|} \subseteq K$.

For example, $\ell_p$-balls for $p \geq 2$ are $(1/p, p)$-uniformly convex w.r.t. $\|\cdot\|_p$ [7, Theorem 2], and $p$-Schatten balls are $(1/p, p)$-uniformly convex w.r.t. the Schatten norm $\|\cdot\|_{S(p)}$ (See [11] and Appendix H for the connection between the uniform convexity of a normed space and the uniform convexity of sets.) Note that $(\kappa, 2)$-uniformly convex sets are $\kappa$-strongly convex.

In this paper, we introduce a new, different characterization of convex bodies.

**Definition 5** (sphere-enclosed sets). Let $K \subseteq \mathbb{R}^d$ be a convex body, $u \in \mathrm{bd}(K)$, and $f \colon K \to \mathbb{R}$. Then, $K$ is $(\rho, u, f)$-*sphere-enclosed* (or simply sphere-enclosed facing $u$) if there exists a sphere of radius $\rho$ that has $u$ on it, encloses $K$, and the gradient of $f$ at point $u$ is directed towards the center of the sphere. That is, there exists a ball $\mathbb{B}(c, \rho)$ with $c \in \mathbb{R}^d$ and

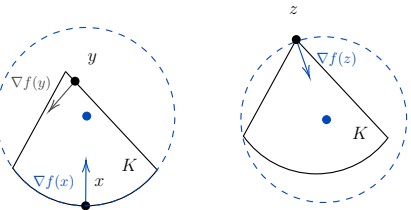

**Figure 1:** Examples of sphere-enclosed sets.

$\rho > 0$ satisfying (i) $u \in \mathrm{bd}(\mathbb{B}(c, \rho))$, (ii) $K \subseteq \mathbb{B}(c, \rho)$, and (iii) there exists $r_0 > 0$ such that $u + r_0 \nabla f(u) = c$.[2]

One might think that the sphere-enclosed condition is complicated but Condition (iii) in Definition 5 is only for the case when $x_\star$ is at the corner of $K$. Figure 1 shows examples of sphere-enclosed sets. The area enclosed by the solid black lines is the convex body $K$. In the left figure, we can see that $K$ is sphere-enclosed facing $x$ (the red dotted line is the minimum sphere facing $x$), but $K$ is not sphere-enclosed facing $y$. In the right figure, we can see that $K$ is sphere-enclosed facing $z$ (the blue dotted line is the minimum sphere facing $z$ for $K$). Note that the notion of sphere-enclosedness is a local property defined for each point of the boundary of convex bodies, in contrast to the definition of strong convexity. In the next section, we will see that we can achieve a logarithmic regret if $K$ is sphere-enclosed facing at optimal decision $x_\star$.

### 2.3.2 Existing lower bound

Here, we discuss a lower bound when exploiting the curvature of feasible sets. For $\lambda \in (0, 1)$, let $W_\lambda = \{(x, y) \in \mathbb{R}^2 : x^2 + y^2/\lambda^2 \leq 1\}$ be an ellipsoid with principal curvature $\lambda$. From [9, Proposition 4], ellipsoid $W_\lambda$ is $\lambda$-strongly convex w.r.t. $\|\cdot\|_2$. The following lower bound provided in [9, Theorem 9] is for this $W_\lambda$, which matches the upper bound in [9, Theorem 5].

**Theorem 6.** *Consider online linear optimization. Let $\lambda, L \in (0, 1)$ and $K = W_\lambda$. Then, for any algorithm, there exists a sequence of linear loss functions $f_1, \ldots, f_T$ satisfying $f_t(\cdot) = \langle g_t, \cdot \rangle$, $g_1, \ldots, g_T \in \{(1, -L), (-1, -L)\}$, and the growth condition that $\|g_1 + \cdots + g_t\|_2 \geq tL$ for all $t \in [T]$ such that $\mathsf{R}_T \geq \frac{1}{84\sqrt{2}} \frac{1}{\lambda L} \ln T - \delta$ for $\delta = \frac{1}{\lambda L} \left( \frac{2}{1 - e^{\lambda^2 L^2}} + \frac{\pi^2}{108} \right)$.*

In their proof, they use the following sequence of linear functions $f_t(\cdot) = \langle h_t^L, \cdot \rangle$. Let $P$ be a random variable following a Beta distribution, $\mathrm{Beta}(k, k)$, for some $k > 0$. For this $P$, let $(X_t)_{t=1}^T$ be i.i.d. random variables following a Bernoulli distribution with parameter $P$. Then for $L \in (0, 1)$, let $h_t^L = (2X_t - 1, -L)$, which indeed satisfies the growth condition $\|h_1^L + \cdots + h_t^L\|_2 \geq tL$ for all $t \in [T]$. This construction of loss functions will be exploited to prove lower bounds in Section 3, and we will provide a matching upper bound in Corollary 12.

### 2.4 Universal online learning

Our algorithm is based on the results of universal online learning. In the literature, the following regret upper bound is the crux for being adaptivity to the curvature of loss functions:

**Lemma 7.** *Consider online convex optimization. Then, there exists an (efficient) algorithm such that $\sum_{t=1}^T \langle \nabla f_t(x_t), x_t - x_\star \rangle$ is bounded from above by the order of*

$$\min \left\{ c_{\mathsf{sc}} \sqrt{\sum_{t=1}^T \|x_t - x_\star\|_2^2 \ln T} + c_{\mathsf{sc}}' \ln T , \ c_{\mathsf{ec}} \sqrt{\sum_{t=1}^T (\langle \nabla f_t(x_t), x_t - x_\star \rangle)^2 \ln T} + c_{\mathsf{ec}}' \ln T , \ GD\sqrt{Tc_{\mathsf{g}}} \right\},$$

(1)

*where $c_{\mathsf{sc}}, c_{\mathsf{sc}}', c_{\mathsf{ec}}, c_{\mathsf{ec}}', c_{\mathsf{g}} > 0$ are algorithm dependent variables provided in the following.*[3]

For example, upper bound (1) can be achieved by the MetaGrad algorithm with $c_{\mathsf{sc}} = G\sqrt{d}$, $c_{\mathsf{sc}}' = d$, $c_{\mathsf{ec}} = \sqrt{d}$, $c_{\mathsf{ec}}' = d$, and $c_{\mathsf{g}} = \ln \ln T$ [21, 22] and the Maler algorithm with $c_{\mathsf{sc}} = G$, $c_{\mathsf{sc}}' = GD$, $c_{\mathsf{ec}} = \sqrt{d}$, $c_{\mathsf{ec}}' = GD + d$, and $c_{\mathsf{g}} = 1$ [24, Theorem 1]. We will see that our regret bounds depend on $c_{\mathsf{sc}}, c_{\mathsf{sc}}', c_{\mathsf{ec}}, c_{\mathsf{ec}}', c_{\mathsf{g}} > 0$, and one can use any algorithm with bound (1).

## 3 Regret lower bounds

In this section, we construct lower bounds that align with the assumptions of our regret bounds. Considering a sequence of loss functions to construct the lower bound in Theorem 6, we can immediately obtain the following lower bound.

---

[2]We will see that the third condition that the gradient of $f$ at point $u$ is directed towards the center of the sphere necessitates careful consideration when optimal decision $x_\star$ is on corners of feasible sets.

[3]The subscripts sc and ec in $c_{\mathsf{sc}}$ and $c_{\mathsf{ec}}$ are the abbreviations of strongly-convex and exp-concave.

**Theorem 8.** *Consider online linear optimization. Let $\lambda, L \in (0, 1)$ and $K = W_\lambda$. Then, for any algorithm, there exists a stochastic sequence of loss functions $f_1, \ldots, f_T$ satisfying $f_t(\cdot) = \langle g_t, \cdot \rangle$, $g_1, \ldots, g_T \in \{(1, -L), (-1, -L)\}$, and $\|\nabla f^\circ(x_\star)\|_2 = L$ such that $\mathsf{R}_T \geq \frac{1}{84\sqrt{2}} \frac{1}{\lambda \|\nabla f^\circ(x_\star)\|_2} \ln T - \delta$, where $\delta$ is defined in Theorem 6.*

*Proof.* Consider the sequence of loss vectors $h_1^L, \ldots, h_T^L$ after Theorem 6 and let $f_t(\cdot) = \langle h_t^L, \cdot \rangle$ for all $t \in [T]$. For this sequence of $(f_t)_{t=1}^T$, it holds that $\|\nabla f^\circ(x_t)\|_2 = \|\mathbb{E}[h_t^L]\|_2 = \|(0, -L)\|_2 = L \neq 0$, which completes the proof. $\qquad\square$

With this lower bound, we have the following lower bound for corrupted stochastic environments.

**Theorem 9.** *Consider online linear optimization. Let $\lambda, L \in (0, 1)$ and $K = W_\lambda$. Suppose that $T \geq C/(\lambda L)^2$ and $C \geq 1/(\lambda L)$. Then, for any algorithm, there exists a corrupted stochastic environment with corruption level at most $C \geq 0$ satisfying $\|\nabla f^\circ(x_\star)\|_2 = L$ such that*

$$\mathsf{R}_T \geq \frac{1}{168\sqrt{2}} \left( \frac{1}{\lambda \|\nabla f^\circ(x_t)\|_2} + \sqrt{\frac{C}{\lambda \|\nabla f^\circ(x_t)\|_2}} \right) \sqrt{\ln\left( \frac{C}{\lambda \|\nabla f^\circ(x_t)\|_2} \right)} - \delta,$$

*where $\delta$ is defined in Theorem 6.*

The assumption that $T \geq C/(\lambda L)^2$ makes some sense since the construction of this lower bound relies on Theorem 8, and if the assumption does not hold then the lower bound becomes vacuous.

*Proof.* We will construct $(f_t)_{t=1}^T$ in a corrupted stochastic environment, where $(f_t)_t$ are generated so that $\tilde{f}_t(\cdot) = \langle \tilde{g}_t, \cdot \rangle$ with $\tilde{g}_1, \ldots, \tilde{g}_T$ following a distribution $\mathcal{D}$ and $f_t$ is a corrupted function of $\tilde{f}_t$.

We first note that we have $T \geq C/(\lambda L) \geq 1/(\lambda L)^2$. Define $\widehat{L} > 0$ such that $\lambda \widehat{L} = \sqrt{\lambda L/C}$. Note that since $C \geq 1/(\lambda L)$, we have $\lambda \widehat{L} \leq \lambda L$, implying that $\widehat{L} \in (0, 1)$. We also define $\tau := \lceil 1/(\lambda \widehat{L})^2 \rceil = \lceil C/(\lambda L) \rceil \leq T$, which follows from $\lambda \widehat{L} = \sqrt{\lambda L/C}$ and $T \geq C/(\lambda L)$.

With these definitions, we then consider the following corrupted stochastic environments:

- For $t \in \{1, \ldots, \tau\}$, define $\tilde{f}_t$ by $\tilde{f}_t(\cdot) = \langle \tilde{g}_t, \cdot \rangle$ for $\tilde{g}_t = h_t^L$, where $h_t^L$ is defined after Theorem 8, and define loss function $f_t$ by $f_t(\cdot) = \langle g_t, \cdot \rangle$ with $g_t = h_t^{\widehat{L}}$.
- For $t \in \{\tau + 1, \ldots, T\}$, let $\tilde{f}_t(\cdot) = f_t(\cdot) = \langle g_t, \cdot \rangle$ with $g_t = h_t^L$, where there is no corruption.

In fact, the corruption level of this environment is bounded by $C$ since $\sum_{t=1}^T \mathbb{E}[\|f_t - \tilde{f}_t\|_\infty] = \sum_{t=1}^\tau \mathbb{E}[\sup_{x \in K} |\langle g_t - \tilde{g}_t, x \rangle|] \leq \tau |L - \widehat{L}| \lambda \leq \lceil C/(\lambda L) \rceil \cdot |L - \widehat{L}| \lambda \leq C$, where in the first inequality we used the fact that the first elements of $g_t$ and $\tilde{g}_t$ are the same and that $K = W_\lambda$ and in the second inequality we used $L \geq \widehat{L} > 0$. This implies that the sequence of $(f_t)_{t=1}^T$ is a corrupted stochastic environment with the corruption level at most $C$.

Hence, from Theorem 8 with $\widehat{L} \in (0, 1)$, $\lambda \widehat{L} = \sqrt{\lambda L/C}$, and the definition of $\tau$, the regret is bounded from below as $\mathsf{R}_T \geq \frac{1}{84\sqrt{2}} \frac{1}{\lambda \widehat{L}} \ln \tau - \delta \geq \frac{1}{84\sqrt{2}} \sqrt{\frac{C}{\lambda L}} \ln\left(\frac{C}{\lambda L}\right) - \delta \geq \frac{1}{84\sqrt{2}} \frac{1}{\lambda L} \ln\left(\frac{C}{\lambda L}\right) - \delta$. Taking the average of the last two inequalities completes the proof. $\qquad\square$

Note that our lower bounds are not for general sphere-enclosed feasible sets, and establishing a new lower bound is important future work.

## 4 Regret upper bounds

In this section, we provide regret upper bounds that nearly match the lower bounds in Section 3, by the universal online learning framework, whose regret is bounded as (1). Note that this section works with convex loss functions.

### 4.1 Regret bounds in stochastic environments

We provide logarithmic regret for stochastic environments. Define the ball $B_\gamma^K \subseteq \mathbb{R}^d$ for $\gamma > 0$ by

$$B_\gamma^K = \mathbb{B}\left(x_\star + \frac{1}{2\gamma}\nabla f^\circ(x_\star), \frac{1}{2\gamma}\|\nabla f^\circ(x_\star)\|_2\right).$$

By the definition, we have $x_\star \in \mathrm{bd}(B_\gamma^K)$. See Figure 2.

**Remark 1.** The ball $B_\gamma^K$ is determined in the following manner. We will see in the following proof that the inequality $\langle \nabla f^\circ(x_\star), x - x_\star \rangle \geq \gamma\|x - x_\star\|_2^2$ that holds for some $\gamma > 0$ and any action $x$ plays a key role in proving a logarithmic regret. This inequality is equivalent to $\|x - (x_\star + \frac{1}{2\gamma}\nabla f^\circ(x_\star))\|_2 \leq \frac{1}{2\gamma}\|\nabla f^\circ(x_\star)\|_2$ and we define $B_\gamma^K$ as the set of all $x \in \mathbb{R}^d$ satisfying this inequality.

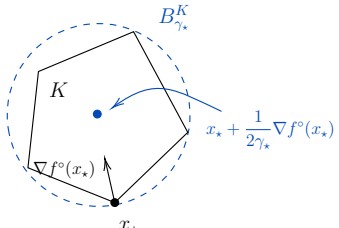

**Figure 2:** The region enclosed by the black solid line is a feasible set $K$ and the red dotted line $B_{\gamma_\star}^K$ is the smallest sphere enclosing $K$ and facing $x_\star$.

Using this $B_\gamma^K$, we let $\gamma_\star = \sup\{\gamma \geq 0 \colon K \subseteq B_\gamma^K\}$. Then, we can prove the following theorem.

**Theorem 10.** *Consider online convex optimization in stochastic environments, where the optimal decision is $x_\star = \arg\min_{x \in K} f^\circ(x)$. Suppose that $K$ is $(\rho, x_\star, f^\circ)$-sphere-enclosed and that $\nabla f^\circ(x_\star) \neq 0$. Then, any algorithm with the bound* (1) *achieves*

$$\mathsf{R}_T = O\left(\frac{c_{\mathsf{sc}}^2}{\gamma_\star}\ln T + c'_{\mathsf{sc}}\ln T\right) = O\left(\frac{c_{\mathsf{sc}}^2 \rho}{\|\nabla f^\circ(x_\star)\|_2}\ln T + c'_{\mathsf{sc}}\ln T\right).$$

The assumption that $K$ is sphere-enclosed around $x_\star$ is satisfied for many typical feasible sets. It holds if the feasible set $K$ is a ball, or a polytope with a mild condition on $\nabla f^\circ(x_\star)$. Specifically, the condition $\nabla f^\circ(x_\star) \in \mathrm{int}(-N_K(x_\star))$ is sufficient to ensure that the feasible $K$ is sphere-enclosed around $x_\star$, where $-N_K(x_\star) = \{g \in \mathbb{R}^d \colon \langle g, x - x_\star \rangle \geq 0 \, \forall x \in K\}$ is the *negative* normal cone. This condition is mild since, from the optimality condition of $x_\star$, we have $\nabla f^\circ(x_\star) \in -N_K(x_\star)$. Hence the undesirable case is restricted to $\nabla f^\circ(x_\star) \in \mathrm{bd}(-N_K(x_\star))$ (see Figure 3 for an example). One might think that the assumption that $x^*$ is on

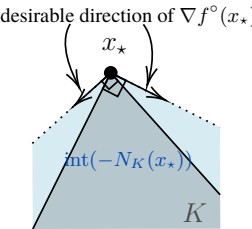

**Figure 3:** An example of an undesirable direction of $\nabla f^\circ(x_\star)$.

the boundary of $K$ is restrictive, but for example, when the loss functions are linear, the minimizer is indeed on the boundary of the feasible set. We will see in Corollary 12 that this upper bound matches the lower bound in Theorem 8 in the environment used to construct the lower bound.

*Proof.* The regret is bounded from below by $\mathsf{R}_T = \mathbb{E}\left[\sum_{t=1}^{T}(f^\circ(x_t) - f^\circ(x_\star))\right] \geq \mathbb{E}\left[\sum_{t=1}^{T}\langle \nabla f^\circ(x_\star), x_t - x_\star \rangle\right] \geq \mathbb{E}\left[\sum_{t=1}^{T}\gamma_\star\|x_t - x_\star\|_2^2\right]$, where the first inequality follows from the convexity of $f^\circ$, and the last inequality follows from $x_t \in K \subseteq B_{\gamma_\star}^K$ and the definition of $\gamma_\star$. By combining this inequality with inequality (1), the regret is bounded as $\mathsf{R}_T \leq \mathbb{E}\left[\sum_{t=1}^{T}\langle \nabla f_t(x_t), x_t - x_\star \rangle\right] = O\left(c_{\mathsf{sc}}\sqrt{\frac{\mathsf{R}_T}{\gamma_\star}\ln T} + c'_{\mathsf{sc}}\ln T\right)$. Solving this inequation w.r.t. $\mathsf{R}_T$, we get $\mathsf{R}_T = O\left(\frac{c_{\mathsf{sc}}^2}{\gamma_\star}\ln T + c'_{\mathsf{sc}}\ln T\right)$. Observing that $\frac{1}{2\gamma_\star}\|\nabla f^\circ(x_\star)\|_2 = \rho$, which holds from the assumption that $K$ is $(\rho, x_\star, f^\circ)$-sphere-enclosed, we complete the proof. $\square$

The advantages of the regret bound in Theorem 10 compared to the existing upper bounds are the following: (i) The logarithmic regret can be achieved as long as the boundary of $K$ is curved around the optimal decision $x_\star$ or $x_\star$ is on corners (see Figure 1), while the existing analysis requires strong convexity over the entire feasible set $K$. (ii) While the existing analysis only considers linear loss functions, our approach can handle convex loss functions and thus the curvature of loss functions (*e.g.*, strong convexity or exp-concavity) can be simultaneously exploited (see Section 4.3). (iii) Even if the growth assumptions on loss vectors $g_1, \ldots, g_T$ are not satisfied, the $O(\sqrt{T})$ regret upper bound can be achieved in adversarial environments, while the existing approach, FTL, can suffer $\Omega(T)$ regret.

A limitation of the proposed approach is that it assumes stochastic environments. However, our approach at least guarantees an $O(\sqrt{T})$ bound in the (fully) adversarial environments, where the growth assumption needed for FTL to achieve the fast rates is not satisfied, and as we will see in the following section, we can achieve the fast rates also in corrupted stochastic environments.

For the assumption on loss vectors, the existing studies consider the following assumptions on loss vectors $g_t$: There exists $L > 0$ such that $\|g_1 + \cdots + g_t\|_2 \geq tL$ for all $t \in [T]$, or $g_t \leq 0$ for all $t \in [T]$. These assumptions cannot be directly comparable with our assumption that $\nabla f^\circ(x_\star) \neq 0$. Note that the assumption that $\inf_{x \in K} \|\nabla f^\circ(x)\| > 0$ is standard in the literature of offline optimization, when deriving the fast convergence rate, see [4, 5, 14] and discussion in [6] for details.

Extending the sphere-enclosed condition to an arbitrary norm is challenging because the sphere-enclosed condition leverages the fact that the enclosing ball is an Euclidean ball. However, we will see in Section 4.4 that fast rates for uniformly convex sets can be achieved for general norms (Theorem 15).

**Remark 2.** The sphere-enclosed condition is similar to the Bernstein condition investigated by Koolen et al. [13]. These conditions are different for general convex loss functions; however, when the loss functions are linear, the sphere-enclosed condition implies the Bernstein condition, allowing us to apply their analysis in this case. Thus, our research can also be viewed as identifying a new condition that satisfies the Bernstein condition. Still, our analysis is more general in the sense that we can deal with general convex loss functions. See Appendix I for a detailed comparison between the sphere-enclosed condition and the Bernstein condition.

**Tightness of regret upper bound in Theorem 10**   In the remainder of this subsection, we investigate the tightness of the regret upper bound in Theorem 10. To see the tightness of our regret bound, we consider the case when $K$ is an ellipsoid. The following proposition implies that the regret upper bound in Theorem 10 matches the lower bound in Theorem 8.

**Proposition 11.** *For $\lambda \in (0, 1)$, let $W_\lambda$ be the ellipsoid defined in Section 2.3.2. Then, its minimum enclosing sphere $S$ such that $(0, -\lambda) \in S$ is $S = \{(x, y) \in \mathbb{R}^2 : x^2 + [y - (1 - \lambda^2)/\lambda]^2 = 1/\lambda^2\}$.*

The proof of Proposition 11 is deferred to Appendix B. This result immediately implies the following:

**Corollary 12.** *Let $K$ be $W_\lambda$ and $x^* = (0, -\lambda)$. Under the same assumption as in Theorem 10, in the environment considered in the construction of the lower bound in Theorem 6, the algorithm in [24] achieves $\mathsf{R}_T = O\left(\frac{1}{\lambda L} \ln T + GD \ln T\right)$.*

This upper bound matches the lower bound in Theorem 8 up to the additive $GD \ln T$ factor.

*Proof.* From Proposition 11 and the fact that Euclidean ball with radius $r$ is $1/r$ strongly convex w.r.t. $\|\cdot\|_2$, we have $\rho = 1/\lambda$. This proposition with Theorem 10 gives the desired bound. $\qquad \square$

The upper bound in Theorem 10 is applicable when $K$ is a polytope. We will see that our approach work also in the corrupted stochastic environment in Section 4.2, and to our knowledge, this is the first upper bound that achieves fast rates when the feasible set is a polytope in non-stochastic environments. A further discussion regarding the case when $K$ is a polytope can be found in Appendix C.

## 4.2   Regret bounds in corrupted stochastic environments

Another advantage of our approach is that it can achieve nearly optimal regret upper bounds even in corrupted stochastic environments. For $\gamma > 0$, we define ball $\widetilde{B}_\gamma^K \subseteq \mathbb{R}^d$ by $\widetilde{B}_\gamma^K = \mathbb{B}\big(\widetilde{x}_\star + \frac{1}{2\gamma}\nabla \widetilde{f}^\circ(\widetilde{x}_\star), \frac{1}{2\gamma}\big\|\nabla \widetilde{f}^\circ(\widetilde{x}_\star)\big\|_2\big)$, which is defined in the same manner as $B_\gamma^K$. For this $\widetilde{B}_\gamma^K$, we let $\widetilde{\gamma}_\star = \sup\{\gamma \geq 0 : K \subseteq \widetilde{B}_\gamma^K\}$. Then, we can prove the following regret upper bound.

**Theorem 13.** *Consider online convex optimization in corrupted stochastic environments with corruption level at most $C$, where $\widetilde{x}_\star = \arg\min_{x \in K} \widetilde{f}^\circ(x)$. Suppose $K$ is $(\rho, \widetilde{x}_\star, \widetilde{f}^\circ)$-sphere enclosed and $\nabla \widetilde{f}^\circ(\widetilde{x}_\star) \neq 0$. Then, any algorithm with the bound* (1) *achieves*

$$\mathsf{R}_T = O\left(\frac{c_{\mathsf{sc}}^2}{\widetilde{\gamma}_\star} \ln T + \sqrt{\frac{Cc_{\mathsf{sc}}^2}{\widetilde{\gamma}_\star} \ln T} + c_{\mathsf{sc}}' \ln T\right) = O\left(\frac{c_{\mathsf{sc}}^2 \rho}{\|\nabla \widetilde{f}^\circ(\widetilde{x}_\star)\|_2} \ln T + \sqrt{\frac{Cc_{\mathsf{sc}}^2 \rho}{\|\nabla \widetilde{f}^\circ(\widetilde{x}_\star)\|_2} \ln T} + c_{\mathsf{sc}}' \ln T\right).$$

The proof of Theorem 13 can be found in Appendix D. One can see that this upper bound matches the lower bound in Theorem 9 up to logarithmic factors. Note that all upper bounds provided in this study can be extended following the same line as the proof of Theorem 13.

### 4.3 Exploiting the curvature of loss functions and feasible set simultaneously

One of the advantages of directly solving OCO over reducing to OLO is that we can obtain upper bounds that can simultaneously exploit the curvature of feasible sets and loss functions:

**Theorem 14.** *Suppose that the same assumption as in Theorem 10 holds. If $f_1, \ldots, f_T$ are $\alpha$-strongly convex w.r.t. a norm $\|\cdot\|$, then $R_T = O\left(\frac{c_{\mathsf{sc}}^2}{\gamma_\star + \alpha/\xi} \ln T + c_{\mathsf{sc}}' \ln T\right)$. If $f_1, \ldots, f_T$ are $\beta$-exp-concave, then $R_T = O\left(\min\left\{\frac{c_{\mathsf{sc}}^2}{\gamma_\star}, \frac{c_{\mathsf{ec}}^2}{\beta' + \gamma_\star/G^2}\right\} \ln T + \max\{c_{\mathsf{sc}}', c_{\mathsf{ec}}'\} \ln T\right)$ for $\beta' \leq \frac{1}{2}\min\left\{\frac{1}{4GD}, \beta\right\}$.*

The proof can be found in Appendix E. Theorem 14 implies that one can simultaneously exploit the curvature of feasible sets and loss functions.

### 4.4 Matching regret upper bound for uniformly convex sets

Here, we prove that a regret bound smaller than $O(\sqrt{T})$ can be achieved when $K$ is uniformly convex. This can be proven by a similar argument using the idea of exploiting the lower bound, as in the proof for sphere-enclosed sets. For uniformly convex feasible sets, we can prove the following theorem.

**Theorem 15.** *Consider online convex optimization in stochastic environments, where the optimal decision is $x_\star = \arg\min_{x \in K} f^\circ(x)$. Suppose that $K$ is $(\kappa, q)$-uniformly convex w.r.t. a norm $\|\cdot\|$ for some $q \geq 2$ and that $\nabla f^\circ(x_\star) \neq 0$. Then, any algorithm with bound (1) achieves*

$$R_T = O\left(\frac{(\xi c_{\mathsf{sc}})^{\frac{q}{q-1}}}{(q\kappa\|\nabla f^\circ(x_\star)\|_\star)^{\frac{1}{q-1}}} T^{\frac{q-2}{2(q-1)}} (\ln T)^{\frac{q}{2(q-1)}} + c_{\mathsf{sc}}' \ln T\right).$$

*In particular, when $K$ is $\lambda$-strongly convex w.r.t. $\|\cdot\|$, $R_T = O\left(\frac{\xi c_{\mathsf{sc}}}{\lambda\|\nabla f^\circ(x_\star)\|_\star} \ln T + c_{\mathsf{sc}}' \ln T\right)$.*

The dependence on $T$ in this bound is $O\left(T^{\frac{q-2}{2(q-1)}} (\ln T)^{\frac{q}{2(q-1)}}\right)$, which becomes $O(\ln T)$ when $q = 2$ and $\tilde{O}(\sqrt{T})$ when $q \to \infty$, and thus interpolates between the bound over the strongly convex sets and non-curved feasible sets. This is strictly better than the $O\left(T^{\frac{q-2}{q-1}}\right)$ bound in [11]; their regret upper bound is better than $O(\sqrt{T})$ only when $q \in (2,3)$. Notably, our bound matches the existing lower bound of $\Omega\left(T^{\frac{q-2}{2(q-1)}}\right)$ proven for a stochastic environment with $d = 2$ [2, Theorem C.1]. For example, when $K$ is $\ell_p$-ball, by $\|x\|_2 \leq d^{\frac{1}{2} - \frac{1}{p}}\|x\|_p$ for any $x \in \mathbb{R}^d$ and $p > 2$, the regret is bounded as $R_T = O\left(\frac{(\xi c_{\mathsf{sc}})^{\frac{p}{p-1}} d^{\frac{p-2}{2(p-1)}}}{(\|\nabla f^\circ(x_\star)\|_\star)^{\frac{1}{p-1}}} T^{\frac{p-2}{2(p-1)}} (\ln T)^{\frac{p}{2(p-1)}} + c_{\mathsf{sc}}' \ln T\right)$.

It is worth noting that the result of Theorem 15 corresponds to the result for sphere-enclosed sets when $q = 2$ and $\|\cdot\|$ is the Euclidean norm. Additionally, Theorem 15 does not require the feasible set $K$ to be globally "curved." In fact, the proof of Theorem 15 only uses the inequality $\langle \nabla f^\circ(x_\star), x - x_\star \rangle \geq \frac{\kappa}{4}\|x - x_\star\|^q \cdot \|\nabla f^\circ(x_\star)\|_\star$ for all $x \in K$, which describes the local curvature around the optimal solution $x_\star$.

Before proving Theorem 15, we present the following lemma, which provides a characterization of uniformly convex sets. This directly follows from the definition in Definition 4:

**Lemma 16.** *Suppose that a convex body $K$ is $(\kappa, q)$-uniformly convex w.r.t. a norm $\|\cdot\|$ for $\kappa > 0$ and $q \geq 2$. Let $y \in K$, $g \in \mathbb{R}^d$, and $y_\star \in \arg\min_{y' \in K}\langle g, y' \rangle$. Then, $\langle -g, y_\star - y \rangle \geq \frac{\kappa}{4}\|y - y_\star\|^q \cdot \|g\|_\star$.*

The proof can be found in [12, Lemma 2.1], and we include the proof in Appendix F for completeness.

*Proof of Theorem 15.* From $x_\star = \arg\min_{x \in K} f^\circ(x)$ and the first-order optimality condition, $\langle \nabla f^\circ(x_\star), x - x_\star \rangle \geq 0$ for all $x \in K$, which implies that $x_\star = \arg\min_{x \in K}\langle \nabla f^\circ(x_\star), x \rangle$. Hence,

by combining this with Lemma 16 and that $K$ is $(\kappa, q)$-uniformly convex w.r.t. a norm $\|\cdot\|$, we have $\langle \nabla f^\circ(x_\star), x - x_\star \rangle \geq \frac{\kappa}{4} \|x - x_\star\|^q \cdot \|\nabla f^\circ(x_\star)\|_\star$ for all $x \in K$. Using this inequality,

$$\mathsf{R}_T \geq \mathbb{E}\left[\sum_{t=1}^T \langle \nabla f^\circ(x_\star), x_t - x_\star \rangle\right] \geq \frac{\kappa}{4}\|\nabla f^\circ(x_\star)\|_\star \mathbb{E}\left[\sum_{t=1}^T \|x - x_\star\|^q\right]$$

$$\geq \frac{\kappa}{4\xi^q}\|\nabla f^\circ(x_\star)\|_\star \mathbb{E}\left[\sum_{t=1}^T \|x - x_\star\|_2^q\right] \geq \frac{\kappa}{4\xi^q}\|\nabla f^\circ(x_\star)\|_\star T^{1-\frac{q}{2}}\left(\mathbb{E}\left[\sum_{t=1}^T \|x - x_\star\|_2^2\right]\right)^{q/2}, \quad (2)$$

where the first inequality follows from the convexity of $f^\circ$, the second inequality by $\|x\|_2 \leq \xi\|x\|$ for any $x \in \mathbb{R}^d$, and the last inequality by Jensen's inequality and the fact that $x \mapsto x^{q/2}$ is convex for $q \geq 2$. Combining (1) and (2), we can bound the regret as
$\mathsf{R}_T = 2\mathsf{R}_T - \mathsf{R}_T$

$$= O\left(c_{\mathsf{sc}}\sqrt{\mathbb{E}\left[\sum_{t=1}^T \|x_t - x_\star\|_2^2\right]\ln T} + c'_{\mathsf{sc}}\ln T\right) - \frac{\kappa}{4\xi^q}\|\nabla f^\circ(x_\star)\|_\star T^{1-\frac{q}{2}}\left(\mathbb{E}\left[\sum_{t=1}^T \|x - x_\star\|_2^2\right]\right)^{q/2}$$

$$= O\left(\frac{(\xi c_{\mathsf{sc}})^{\frac{q}{q-1}}}{(q\kappa\|\nabla f^\circ(x_\star)\|_\star)^{\frac{1}{q-1}}}T^{\frac{q-2}{2(q-1)}}(\ln T)^{\frac{q}{2(q-1)}} + c'_{\mathsf{sc}}\ln T\right),$$

where in the last line we used the inequality $a\sqrt{x} - bx^{q/2} \leq a^{\frac{q}{q-1}}/(qb)^{\frac{1}{q-1}}$ that holds for $a, b, x > 0$ and $q \geq 2$. This completes the proof. $\qquad\square$

The above analysis can be extended to corrupted stochastic environments in a straightforward manner:

**Theorem 17.** *Consider online convex optimization in corrupted stochastic environments with corruption level $C$, where the optimal decision is $\widetilde{x}_\star = \arg\min_{x \in K}\widetilde{f}^\circ(x)$. Suppose that $K$ is $(\kappa, q)$-uniformly convex w.r.t. a norm $\|\cdot\|$ for some $q \geq 2$ and that $\nabla \widetilde{f}^\circ(\widetilde{x}_\star) \neq 0$. Then, any algorithm with the bound* (1) *achieves*

$$\mathsf{R}_T = O\left(\mathcal{R} + C^{\frac{1}{q}}\mathcal{R}^{\frac{q-1}{q}} + c'_{\mathsf{sc}}\ln T\right) \quad \text{for} \quad \mathcal{R} = \frac{(\xi c_{\mathsf{sc}})^{\frac{q}{q-1}}}{(q\kappa\|\nabla\widetilde{f}^\circ(\widetilde{x}_\star)\|_\star)^{\frac{1}{q-1}}}T^{\frac{q-2}{2(q-1)}}(\ln T)^{\frac{q}{2(q-1)}}.$$

*When $K$ is $\lambda$-strongly convex w.r.t. $\|\cdot\|$, $\mathsf{R}_T = O\left(\frac{\xi c_{\mathsf{sc}}}{\lambda\|\nabla\widetilde{f}^\circ(\widetilde{x}_\star)\|_\star}\ln T + \sqrt{\frac{C\xi c_{\mathsf{sc}}}{\lambda\|\nabla\widetilde{f}^\circ(\widetilde{x}_\star)\|_\star}\ln T} + c'_{\mathsf{sc}}\ln T\right).$*

The proof can be found in Appendix G. When $q = 2$, the dependence on the corruption level $C$ is the same as that in Theorem 13.

## 5   Conclusion

In this work, we introduce a new curvature condition for achieving fast rates in online convex optimization. Under this condition, we developed a new analysis to achieve fast rates by exploiting the curvature of feasible sets. In particular, by the algorithm adaptive to the curvature of loss functions, we proved an $O(\rho \ln T)$ regret bound for $(\rho, x_\star, f^\circ)$-sphere enclosed feasible sets. There are several advantages of our approach: it can exploit the curvature of loss functions, can achieve the $O(\ln T)$ regret bound only with local curvature properties, and can work robustly even in environments where loss vectors do not satisfy the ideal conditions. Notably, following a similar analysis, we proved the matching regret upper bound for uniformly convex feasible sets, which include strongly convex sets and $\ell_p$-balls for $p \in [2, \infty)$ as special cases. This regret bound interpolates the $O(\ln T)$ regret over strongly convex sets and the $O(\sqrt{T})$ regret over non-curved sets.

## Acknowledgments and Disclosure of Funding

The authors would like to express their gratitude to Taiji Suzuki for the insightful discussions that led to the idea of exploiting the curvature of feasible sets in online learning. The authors would also like to grateful to the anonymous reviewers for their insightful feedback and constructive suggestions, which have helped to significantly improve the manuscript. TT was supported by JST ACT-X Grant Number JPMJAX210E and JSPS KAKENHI Grant Number JP24K23852.

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

## A   Additional related work

This appendix discusses the additional related work.

**Fast rates on strongly or uniformly convex sets**   Exploiting the curvature of feasible sets has been considered in OLO. In addition to the previously discussed studies [9, 11, 17], in online learning with a hint, where a context $m_t$ satisfying $m_t^\top x_t \geq c\|x_t\|_2^2$ is given every round, we can achieve an $O(\frac{1}{c}\ln T)$ regret [3], which was further extended in [2] with the lower bound for uniformly convex sets. The curvature was also exploited for improving a regret upper bound and reducing the number of linear optimization oracle calls in constructing Frank-Wolfe-based algorithms [16, 23]. Beyond the scope of online learning, exploiting the curvature of feasible sets has been investigated also in offline optimization, where the goal is to solve $\min_{x \in K} f(x)$ for a given smooth convex function $f$ [4, 5, 6, 11, 14].

**Fast rates on curved loss functions**   One classical and seminal work to exploit the curvature of loss functions is by Hazan et al. [8]. Our approach is based on the results of universal online learning, the motivation of which is to be adaptive to parameters of loss curvature without knowing them. This line of investigation was initiated by van Erven and Koolen [21], van Erven et al. [22], who establish an algorithm, MetaGrad, that achieves an $O(\frac{d}{\beta}\ln T)$ regret bound if loss functions are $\beta$-exp-concave and an $O(\sqrt{T \ln \ln T})$ bound otherwise, without knowing the curvature of loss functions. The underlying idea is to run several experts in parallel with different curvature parameters, and then another expert algorithm integrates their results to choose decisions. This algorithm was later extended to achieve an $O(\frac{1}{\alpha}\ln T)$ regret bound when the loss functions are $\alpha$-strongly convex [24]. Roughly speaking, this was made possible by considering MetaGrad with additional experts of OGD with a learning rate of $\Theta(1/t)$. They provided a bound of $\sum_{t=1}^T \langle \nabla f_t(x_t), x_t - x_\star \rangle = O(G\sqrt{\sum_{t=1}^T \|x_t - x_\star\|_2^2 \ln T} + GD \ln T)$ for $D = \max_{x,y \in K} \|x - y\|_2$. This universal online learning framework has been extended to a simpler form [26] and to a form with the path-length bound [25]. It is worth noting that these algorithms are efficient since the number of experts is at most $O(\ln T)$.

**Intermediate environments in OCO**   The corrupted stochastic environments considered in this paper are similar to the formulations investigated in the context of expert problems and multi-armed bandit problems [10, 15], and this environment is also referred to as stochastic environments with adversarial corruptions. Sachs et al. [19, 20] considered a more general environment, the stochastically extended adversary (SEA) model. It would be important future work to extend our results to the SEA model. Note that they do not consider the curvature of feasible sets.

## B   Proof of Proposition 11

*Proof.* Since $K$ is $W_\lambda$ and $x_\star = (0, -\lambda)$, the optimization problem we need to solve is formulated as follows:

$$\underset{r>0,\, c>0}{\text{minimize}} \quad r^2 \quad \text{subject to} \quad \sup_{(x,y)\in K} \left\| \begin{pmatrix} x \\ y \end{pmatrix} - \begin{pmatrix} 0 \\ c \end{pmatrix} \right\|^2 \leq r^2, \; \mathbb{B}((0,c),r) \text{ is tangent to } x_\star. \quad (3)$$

From geometric observations, we have $c - r = -\lambda$. Hence, the optimization problem (3) can be rewritten as

$$\underset{c \in \mathbb{R}}{\text{minimize}} \quad (c + \lambda)^2 \quad \text{subject to} \quad \sup_{(x,y)\in K} \{x^2 + y^2 - 2cy\} \leq 2c\lambda + \lambda^2. \quad (4)$$

In the following, to make the constraint in the optimization problem (4) simpler, we consider the following optimization problem:

$$\underset{(x,y)\in \mathbb{R}^2}{\text{maximize}} \quad x^2 + y^2 - 2cy \quad \text{subject to} \quad x^2 + \frac{y^2}{\lambda^2} \leq 1.$$

By the standard method of Lagrange multiplier, one can compute that the optimal value of this optimization problem is $\max\left\{-2\lambda c + \lambda^2, 2\lambda c + \lambda^2, \frac{1}{\frac{1}{\lambda^2}-1}c^2 + 1\right\}$. Since the inequality

$$\max\left\{-2\lambda c + \lambda^2, 2\lambda c + \lambda^2, \frac{1}{\frac{1}{\lambda^2}-1}c^2 + 1\right\} \le 2\lambda c + \lambda^2$$

only holds when $c = \frac{1-\lambda^2}{\lambda}$, the feasible set of the optimization problem (4) is singleton set $\left\{\frac{1-\lambda^2}{\lambda}\right\}$. Combining this fact with $c - r = -\lambda$, we get the desired result. □

## C   Discussion when feasible set is polytope

Here we discuss the pros/cons of our regret bound against an existing bound when feasible set $K$ is a polytope. The results mentioned in Section 4 mainly focus on the case where the feasible set is curved. However, as one can see from the definition of the sphere-enclosedness, even if the feasible set is polytope or does not have the curvature, a regret upper bound better than $O(\sqrt{T})$ can be achieved.

In the existing study, the following upper bound is known in stochastic OCO over polytope [9, Corollary 11].

**Theorem 18.** *Consider online online linear optimization in stochastic environments with $g^\circ = \mathbb{E}[g_t]$. Assume that $K$ is polytope and $\|g_t\|_\infty \le M$. Further assume that there exsits $r > 0$ such that $\Phi(\cdot) = \max_{x \in K}\langle x, \cdot \rangle$ is differentiable for any $\nu$ such that $\|\nu - g^\circ\| \le r$. Then, the regret of FTL is bounded by $\mathsf{R}_T = O\left(\frac{M^3 D}{r^2}\ln\left(\frac{M^2 d}{r^2}\right)\right)$.*

The comparison of this bound with our regret upper bound is not straightforward. If $\nabla f^\circ(x_\star)$ is not toward the "unfavorable" direction in $K$, then it is trivial that polytope $K$ is $(\rho, x_\star, f^\circ)$-sphere-enclosed for some $\rho$, and thus the regret bound in Theorem 10 can be achieved. When $T$ is large enough, the bound in Theorem 18 is better since it does not depend on $T$. However, their regret upper bound depends on $1/r^2$ and $M^3$, and the relation between them and $1/\|\nabla f^\circ(x_t)\|_2$ is unclear, and our bound can be smaller than their bound. Note that the "unfavorable" direction coincides between these upper bounds when OLO is considered.

While the direct comparison is not straightforward, we would like to emphasize that our regret upper bound, in contrast to their bound, is obtained as a corollary of the general analysis, and our bound inherits all the advantages discussed in Section 4. In particular, while their bound is valid only in stochastic environments, our regret guarantee is valid in stochastic, adversarial, and corrupted stochastic environments.

## D   Proof of Theorem 13

*Proof.* Recalling that $\widetilde{x}_\star = \arg\min_{x \in K} \widetilde{f}^\circ(x)$, we can bound the regret from below as

$$\mathsf{R}_T = \max_{x \in K} \mathbb{E}\left[\sum_{t=1}^{T}(f_t(x_t) - f_t(x))\right]$$

$$= \max_{x \in K}\left\{\mathbb{E}\left[\sum_{t=1}^{T}\left(\tilde{f}_t(x_t) - \tilde{f}_t(x)\right)\right] + \mathbb{E}\left[\sum_{t=1}^{T}\left(\left(f_t(x_t) - \tilde{f}_t(x_t)\right) - \left(f_t(x) - \tilde{f}_t(x)\right)\right)\right]\right\}$$

$$\ge \max_{x \in K} \mathbb{E}\left[\sum_{t=1}^{T}\left(\tilde{f}_t(x_t) - \tilde{f}_t(x)\right)\right] - 2C\,.$$

The first term in the last inequality is further bounded from below as

$$\max_{x \in K} \mathbb{E}\left[\sum_{t=1}^{T}\left(\tilde{f}_t(x_t) - \tilde{f}_t(x)\right)\right] \ge \mathbb{E}\left[\sum_{t=1}^{T}\left(\widetilde{f}^\circ(x_t) - \widetilde{f}^\circ(x_\star)\right)\right]$$

$$\ge \mathbb{E}\left[\sum_{t=1}^{T}\langle\nabla\widetilde{f}^\circ(\widetilde{x}_\star), x_t - x_\star\rangle\right] \ge \mathbb{E}\left[\sum_{t=1}^{T}\widetilde{\gamma}_\star\|x_t - x_\star\|_2^2\right],$$

where the first inequality follows by the definition of $\widetilde{x}_\star$ and the last inequality follows by $x_t \in K \subseteq \tilde{B}_{\gamma_\star}^K$. Combining the above inequalities with (1), we have $\mathsf{R}_T = O\left(c_{\mathsf{sc}}\sqrt{\frac{\mathsf{R}_T + C}{\widetilde{\gamma}_\star}\ln T} + c'_{\mathsf{sc}}\ln T\right)$. Solving this inequation and following the similar analysis as the proof of Theorem 10 complete the proof. $\square$

## E    Proof of Theorem 14

*Proof.* Following the same argument as in the proof of Theorem 10, we have

$$\mathsf{R}_T = 2\mathsf{R}_T - \mathsf{R}_T \leq 2\mathsf{R}_T - \mathbb{E}\left[\sum_{t=1}^{T}\gamma_\star\|x_t - x_\star\|_2^2\right]. \tag{5}$$

From the strong convexity of $f_t$, we also have

$$\mathsf{R}_T \leq \mathbb{E}\left[\sum_{t=1}^{T}\left(\langle\nabla f_t(x_t), x_t - x_\star\rangle - \frac{\alpha}{2}\|x_t - x_\star\|^2\right)\right]$$

$$\leq \mathbb{E}\left[\sum_{t=1}^{T}\left(\langle\nabla f_t(x_t), x_t - x_\star\rangle - \frac{\alpha}{2\xi}\|x_t - x_\star\|_2^2\right)\right]. \tag{6}$$

Plugging (6) in (5) and from Lemma 7 and Jensen's inequality,

$$\mathsf{R}_T = O\left(c_{\mathsf{sc}}\sqrt{\mathbb{E}\left[\sum_{t=1}^{T}\|x_t - x_\star\|_2^2\right]\ln T} + c'_{\mathsf{sc}}\ln T\right) - \left(\frac{\alpha}{2\xi} + \gamma_\star\right)\mathbb{E}\left[\sum_{t=1}^{T}\|x_t - x_\star\|_2^2\right]$$

$$= O\left(\frac{c_{\mathsf{sc}}^2}{\alpha/\xi + \gamma_\star}\ln T + c'_{\mathsf{sc}}\ln T\right),$$

which completes the proof for the strongly convex loss functions.

Next, we consider the case where $f_t$'s are exp-concave. By [8, Lemma 3], the $G$-Lipschitzness and $\beta$-exp-concavity of $f_t$ implies

$$f_t(x_\star) \geq f_t(x_t) + \langle\nabla f_t(x_t), x_\star - x_t\rangle + \frac{\beta'}{2}(\langle\nabla f_t(x_t), x_t - x_\star\rangle)^2$$

for $\beta' \leq \frac{1}{2}\min\{\frac{1}{4GD}, \beta\}$. Using this and Lemma 7 to follow a similar argument as the strongly-convex case, we can bound the regret as

$$\mathsf{R}_T = O\left(\sqrt{\min\left\{c_{\mathsf{sc}}^2\mathbb{E}\left[\sum_{t=1}^{T}\|x_t - x_\star\|_2^2\right], c_{\mathsf{ec}}^2\mathbb{E}\left[\sum_{t=1}^{T}(\langle\nabla f_t(x_t), x_t - x_\star\rangle)^2\right]\right\}\ln T}\right.$$

$$\left. + \max\{c'_{\mathsf{sc}}, c'_{\mathsf{ec}}\}\ln T\right) - \gamma_\star\mathbb{E}\left[\sum_{t=1}^{T}\|x_t - x_\star\|_2^2\right] - \beta'\mathbb{E}\left[\sum_{t=1}^{T}(\langle\nabla f_t(x_t), x_t - x_\star\rangle)^2\right]$$

$$= O\left(\min\left\{\frac{c_{\mathsf{sc}}^2}{\gamma_\star}, \frac{c_{\mathsf{ec}}^2}{\beta' + \gamma_\star/G^2}\right\} + \max\{c'_{\mathsf{sc}}, c'_{\mathsf{ec}}\}\ln T\right),$$

where in the last inequality we used $\|x_t - x_\star\|_2^2 \geq \frac{1}{G^2}\left(\nabla f_t(x_t)^\top(x_t - x_\star)\right)^2$ that holds by the Cauchy–Schwarz inequality. $\square$

## F    Proof of Lemma 16

*Proof.* Since $K$ is $(\kappa, q)$-uniformly convex w.r.t. norm $\|\cdot\|$,

$$\frac{y + y_\star}{2} + \frac{\kappa}{8}\|y - y_\star\|^q \cdot \mathbb{B}_{\|\cdot\|} \subseteq K.$$

Hence, for any $z \in \mathbb{B}_{\|\cdot\|}$, the definition of $y_\star$ implies that

$$\langle g, y_\star \rangle \le \left\langle g, \frac{y + y_\star}{2} + \frac{\kappa}{8}\|y - y_\star\|^q z \right\rangle = \left\langle g, \frac{y + y_\star}{2} \right\rangle + \left\langle g, \frac{\kappa}{8}\|y - y_\star\|^q z \right\rangle.$$

Rearranging the last inequality implies $\langle -g, y_\star - y \rangle \ge \frac{\kappa}{4}\|y - y_\star\|^q \langle -g, z \rangle$. Choosing $z = -g/\|g\| \in \mathbb{B}_{\|\cdot\|}$ completes the proof. $\qquad\square$

# G  Proof of Theorem 17

*Proof.* The regret is bounded from below as

$$\mathsf{R}_T \ge \mathbb{E}\left[\sum_{t=1}^{T} \langle \nabla \widetilde{f}^\circ(\widetilde{x}_\star), x_t - x_\star \rangle\right] - 2C \ge \frac{\kappa}{4\xi^q}\|\nabla \widetilde{f}^\circ(\widetilde{x}_\star)\|_\star T^{1-\frac{q}{2}} \left(\mathbb{E}\left[\sum_{t=1}^{T}\|x - x_\star\|_2^2\right]\right)^{q/2} - 2C, \tag{7}$$

where the first inequality follows from the same argument as in the proof of Theorem 13 in Appendix D and the second inequality from the same argument as in (2). Recall that $\mathcal{R} = \frac{(\xi c_{\mathsf{sc}})^{\frac{q}{q-1}}}{\left(q\kappa\|\nabla\widetilde{f}^\circ(\widetilde{x}_\star)\|_\star\right)^{\frac{1}{q-1}}} T^{\frac{q-2}{2(q-1)}} (\ln T)^{\frac{q}{2(q-1)}}$. Then from (1) and (7), for any $\lambda \in (0, 1]$ we have

$$\mathsf{R}_T = (1 + \lambda)\mathsf{R}_T - \lambda \mathsf{R}_T$$

$$\le (1 + \lambda)c_{\mathsf{sc}}\sqrt{\mathbb{E}\left[\sum_{t=1}^{T}\|x_t - x_\star\|_2^2\right] \ln T} - \frac{\kappa}{4\xi^q}\|\nabla\widetilde{f}^\circ(\widetilde{x}_\star)\|_\star T^{1-\frac{q}{2}}\left(\mathbb{E}\left[\sum_{t=1}^{T}\|x - x_\star\|_2^2\right]\right)^{q/2}$$

$$\quad + 2\lambda C + (1 + \lambda)c'_{\mathsf{sc}} \ln T$$

$$\le \frac{(1 + \lambda)^{\frac{q}{q-1}}}{\lambda^{\frac{1}{q-1}}}\mathcal{R} + 2\lambda C + 2c'_{\mathsf{sc}} \ln T$$

$$\le \frac{4}{\lambda^{\frac{1}{q-1}}} \frac{c_{\mathsf{sc}}^{\frac{q}{q-1}}}{(qz)^{\frac{1}{q-1}}} T^{\frac{q-2}{2(q-1)}} (\ln T)^{\frac{q}{2(q-1)}} + 2\lambda C + 2c'_{\mathsf{sc}} \ln T \tag{8}$$

where in the second inequality we used $a\sqrt{x} - bx^{q/2} \le a^{\frac{q}{q-1}}/(qb)^{\frac{1}{q-1}}$ that holds for $a, b, x > 0$ and $q \ge 2$ and in the last inequality we used $(1+\lambda)^{\frac{q}{q-1}} \le 2^{\frac{q}{q-1}} \le 4$. Choosing $\lambda = \left(\frac{\mathcal{R}}{C+\mathcal{R}}\right)^{\frac{q-1}{q}} \in (0, 1]$ in (8) gives $\mathsf{R}_T = O(\mathcal{R} + C^{\frac{1}{q}}\mathcal{R}^{\frac{q-1}{q}} + c'_{\mathsf{sc}} \ln T)$, which completes the proof. $\qquad\square$

# H  Connection between uniform convexity in Banach space and uniformly convex sets

This appendix discusses the connection between the uniform convexity in Banach space and the uniformly convex sets. While the notion of uniformly convex set was employed in the context of achieving fast rates by exploiting the curvature of feasible sets [11], some papers in the context of online learning with a hint consider *uniformly convex Banach spaces* [2, 3]. This appendix may be useful in making the claims of the main body clearer by clarifying these relationships.

**Uniformly convex space and modulus of uniform convexity**  We start from the definition of the uniform convexity in Banach spaces [18, Definition 4.16].

**Definition 19.** A Banach space $(B, \|\cdot\|)$ is *uniformly convex* if for any $\varepsilon \in (0, 2]$ there exists a $\delta > 0$ such that

$$\forall x, y \in B, \quad \|x\| \le 1, \|y\| \le 1, \|x - y\| \ge \varepsilon \implies \left\|\frac{x + y}{2}\right\| \le 1 - \delta.$$

The *modulus of uniform convexity* $\delta_B(\varepsilon)$ is the best possible $\delta$ for that $\varepsilon$, that is,

$$\delta_B(\varepsilon) = \inf\left\{1 - \left\|\frac{x + y}{2}\right\| : \|x\| \le 1, \|y\| \le 1, \|x - y\| \ge \varepsilon\right\}.$$

Further, we say that $B$ is $(C, q)$-uniformly convex if $\delta_B(\varepsilon) = C\varepsilon^q$.

We say that a Banach space $B$ is uniformly convex if $\delta_B(\varepsilon) > 0$ for any $\varepsilon \in (0, 2]$. The modulus of convexity captures the strength of convexity, and intuitively, if the convexity of the space is large, then any two arbitrarily chosen points on the unit sphere will have their midpoints located deep inside the unit sphere. From this intuition, one can imagine that $\ell_\infty$ space is not uniformly convex. In fact, $x = (1, 1, 1, \dots)$, $y = (-1, 1, 1, \dots)$ satisfies $\|x\|_\infty = \|y\|_\infty = 1$, $\|x - y\|_\infty = 2$ but $1 - \|(x + y)/2\|_\infty = 0$.

**Uniformly convex sets**  Here, we adopt a slightly generalized definition of uniformly convexity.

**Definition 20.** A convex body $K$ is $\gamma_K(\cdot)$-uniformly convex w.r.t. a norm $\|\cdot\|$ if for any $x, y \in K$ and any $\theta \in [0, 1]$, it holds that

$$\theta x + (1 - \theta)y + \theta(1 - \theta)\gamma_K(\|x - y\|) \cdot \mathbb{B}_{\|\cdot\|} \subseteq K \,.$$

In particular, we say that a convex body $K$ is $(\kappa, q)$-uniformly convex w.r.t. a norm $\|\cdot\|$ (or $q$-uniformly convex) if $\gamma_K(r) \geq \kappa r^q$.

**Connection between uniform convexity in Banach space and uniformly convex sets**  It is known that the unit ball on a uniformly convex space is a uniformly convex set and their uniform convexity matches up to a constant factor [11, Lemma 4.2].

**Proposition 21.** *Suppose that a Banach space is uniformly convex with a modulus of convexity $\delta(\cdot)$. Then the unit ball on the Banach space, $\mathbb{B}_{\|\cdot\|}$, is $2\delta(\cdot)$-uniformly convex set w.r.t. $\|\cdot\|$. That is, for any $x, y \in \mathbb{B}_{\|\cdot\|}$ and any $\theta \in [0, 1]$, $\theta x + (1 - \theta)y + \theta(1 - \theta)2\delta(\|x - y\|) \cdot \mathbb{B}_{\|\cdot\|} \subseteq \mathbb{B}_{\|\cdot\|}$.*

In existing studies, there is no explicit discussion on whether the converse of Proposition 21 is true. However, they are necessary to convert the major result known for uniformly convex spaces (*e.g.,* [7, Theorem 2]) into a result for uniformly convex balls. In the following, we show that the converse of Proposition 21 indeed holds for completeness:

**Proposition 22.** *If a ball $K = \mathbb{B}_{\|\cdot\|}$ is $\gamma(\cdot)$-uniformly convex set w.r.t. a norm $\|\cdot\|$, then a Banach space with the norm $\|\cdot\|$ is uniformly convex with a modulus of convexity $\frac{1}{4}\gamma(\cdot)$.*

*Proof.* Since $K$ is $\gamma(\cdot)$-uniformly convex w.r.t. a norm $\|\cdot\|$, it holds for any $x, y \in K$ and $z \in \mathbb{B}_{\|\cdot\|}$ that

$$\frac{x + y}{2} + \frac{1}{4}\gamma(\|x - y\|)z \in K \,.$$

Taking $z = \frac{1}{2}(x + y)/\|\frac{1}{2}(x + y)\| \in \mathbb{B}_{\|\cdot\|}$ implies

$$\frac{x + y}{2} + \frac{1}{4}\gamma(\|x - y\|) \cdot \frac{(x + y)/2}{\|(x + y)/2\|} \in K \,.$$

From this observation, we obtain

$$\left\| \frac{x + y}{2} + \frac{1}{4}\gamma(\|x - y\|) \cdot \frac{(x + y)/2}{\|(x + y)/2\|} \right\| = \left( 1 + \frac{1}{4}\gamma(\|x - y\|)\frac{1}{\|(x + y)/2\|} \right) \left\| \frac{x + y}{2} \right\|$$

$$= \left\| \frac{x + y}{2} \right\| + \frac{1}{4}\gamma(\|x - y\|) \leq 1 \,,$$

where the last inequality follows from the assumption that $K = \mathbb{B}_{\|\cdot\|}$. Hence, for any $x, y \in K = \mathbb{B}_{\|\cdot\|}$, it holds that

$$1 - \left\| \frac{x + y}{2} \right\| \geq \frac{1}{4}\gamma(\|x - y\|) \,,$$

which implies that Banach space with the norm $\|\cdot\|$ is uniformly convex with the modulus of convexity of $\delta(\cdot) = \frac{1}{4}\gamma(\cdot)$. $\qquad\square$

# I  Comparison of sphere-enclosed condition and Bernstein condition

This appendix discusses the relation between the sphere-enclosed condition and the Bernstein condition. We use $\mathbb{E}_t[\,\cdot\,]$ to denote the expectation given $f_1, \dots, f_{t-1}$.

**Bernstein condition** The seminal paper by Koolen et al. [13] provides the following Bernstein condition to obtain fast rates in OCO:

**Definition 23.** In online convex optimization, a sequence of loss functions $(f_t)_{t=1}^T$ satisfies the $(B, \kappa)$-Bernstein condition if 10

$$\mathbb{E}_t\Big[(\langle \nabla f_t(x), x - x_\star \rangle)^2\Big] \leq B \, \mathbb{E}_t[\langle \nabla f_t(x), x - x_\star \rangle]^\kappa \qquad (9)$$

almost surely for all $x \in K$ and $t \in [T]$.

When $\kappa = 1$, this condition is also known as the Massart condition. They proved that if the $(B, \kappa)$-Bernstein condition is satisfied, then the MetaGrad algorithm achieves a regret bound of $\mathsf{R}_T = O\Big((d \ln T)^{\frac{1}{2-\kappa}} T^{\frac{1-\kappa}{2-\kappa}}\Big)$.

**Sphere-enclosed condition** Under the same assumption as in Theorem 10, the $(\rho, x_\star, f^\circ)$-sphere-enclosed condition implies that

$$\langle \nabla f^\circ(x_\star), x - x_\star \rangle \geq \gamma^* \|x - x_\star\|_2^2 = \frac{\|\nabla f^\circ(x_\star)\|_2}{2\rho} \|x - x_\star\|_2^2$$

for any $x \in K$. Rearranging the last inequality gives that

$$\|x - x_\star\|_2^2 \leq \frac{2\rho}{\|\nabla f^\circ(x_\star)\|_2} \langle \nabla f^\circ(x_\star), x - x_\star \rangle \qquad (10)$$

holds for any $x \in K$. When loss functions are stochastic, the last inequality implies

$$\mathbb{E}_t\Big[(\langle \nabla f_t(x), x - x_\star \rangle)^2\Big] \leq G^2 \|x - x_\star\|_2^2 \leq \frac{2G^2\rho}{\|\nabla f^\circ(x_\star)\|_2} \langle \nabla f^\circ(x_\star), x - x_\star \rangle, \qquad (11)$$

where the first inequality follows from the Cauchy–Schwarz inequality and the second inequality follows from (10).

**Comparison between Bernstein condition and sphere-enclosed condition** Comparing (9) and (11), we can see that they look similar but different and there is no direct connection between the sphere-enclosed condition and the Bernstein condition, since the RHS of (9) has $x$ in $\nabla f_t(x)$ while the RHS of (11) has $x_\star$ in $\nabla f^\circ(x_\star)$. However, when loss functions are linear with $g^\circ = \nabla f^\circ(x)$ for all $x \in K$, Eq. (11) is equivalent to

$$\mathbb{E}_t\Big[(\langle \nabla f_t(x), x - x_\star \rangle)^2\Big] \leq \frac{2G^2\rho}{\|g^\circ\|_2} \langle \nabla f^\circ(x_\star), x - x_\star \rangle = \frac{2G^2\rho}{\|g^\circ\|_2} \mathbb{E}_t[\langle \nabla f_t(x_t), x - x_\star \rangle].$$

Therefore, when loss functions are stochastic and linear, the $(\rho, x_\star, f^\circ)$-sphere-enclosed condition implies the $(2G^2\rho/\|g^\circ\|_2, 1)$-Bernstein condition for $g^\circ = \nabla f^\circ(x)$. Hence, in stochastic OLO one can directly apply the result in [13] to obtain fast rates. Still, our analysis deals with general convex loss functions, can also be extended to OCO over uniformly convex sets, and has several advantages as detailed in Section 1.

