# OpenReview forum: "Fast Rates in Stochastic Online Convex Optimization by Exploiting the Curvature of Feasible Sets"
_NeurIPS.cc/2024/Conference — NeurIPS 2024 poster_

### Official Review · Reviewer_BHA1 · 2024-06-23

**Soundness:** 3
**Presentation:** 2
**Contribution:** 3
**Rating:** 6
**Confidence:** 3

**Summary:**

- This paper studies online convex optimisation on feasible sets with curvature conditions. They introduce a new way to exploit the curvature of the feasible set through sphere enclosure. For feasible sets satisfying this condition, they leverage results from universal online learning to show that there exists algorithms whose regret is logarithmic in $T$ (for stochastic environments).
- Compared to prior work, the conditions of curvature on the feasible set are more general, the losses are not restricted to be linear (and conditions such as strong convexity of the losses can be simultaneously exploited) and a growth condition on the gradients is not needed. These results are also extended to stochastic corrupted environments.
- For feasible sets that are uniformly convex, they show that there exists algorithm which can achieve regret bounds interpolating between the $\log T$ rate of strongly-convex sets (q=2) and the $\sqrt{T}$-rate of sets without curvature ($q \rightarrow \infty$) (in stochastic environments) while prior work suffered linear in $T$ regret when $q \rightarrow \infty$.

**Strengths:**

- Sphere-enclosure, a new interesting characterisation of the feasible set is introduced, which allows to obtain $\log T$ regret with improvements with respect to prior work (see Summary) that are clearly explained. Sphere-enclosure captures previously considered sets with curvature as well as some other more general ones. Helpful figures are also provided to help understand and visualise sphere-enclosure.
- The theoretical contributions are strong with upper and matching lower-bounds.
- Exploiting the framework of universal online learning allows proofs that are generally simple to follow.
- For uniformly convex sets, the interpolation of $\log T$ and $\sqrt{T}$ rates closes a gap left by prior work (in stochastic environments).

**Weaknesses:**

- The improvement over FTL is mostly in the stochastic environment. In Theorem 13, if we consider the fully adversarial environment ($C=T$) for example, then the regret is $\sqrt{T}$ and the $\log T$ results for FTL are not improved on. Do you believe this is a fault in the analysis or that something beyond the universal online learning conditions are required ?
- Many of the results work with the $\ell_2$-norm and it is not clear what the results would be when working with respect to an arbitrary norm $||\cdot||$. In particular, it would be nice to at least have a discussion of if/how these results could be extended to arbitrary norms. It also seems that considering the euclidean ball in Definition 5 means that $\ell_p$-balls for $p > 2$ are not sphere-enclosed sets. It would be good to have a discussion of the relationship between uniformly convex sets and sphere-enclosed sets as well as a discussion on how the results from Section 4.1 link to the results in Section 4.4.
- The generality of the assumption of sphere-enclosure of the feasible set is well motivated (what is the mild condition on $\nabla f^{\circ}(x_\star)$ in line 231 ?). But the assumption that $x^\star$ is on the boundary of $K$ could be discussed further. In particular, it is only justified for linear losses. Perhaps a slightly more developed discussion of this could help (or/and more pointers to the literature). Moreover, it would be nice to have an explanation of why this assumption is needed - is it just to guarantee $||\nabla f^{\circ}(x_\star)||_2 \neq 0$?

Some sections are slightly unclear and a bit confusing:
- In the statement of Theorem 9, it is stated that $T \geq C/(\lambda L)^2$ (which is also stated in line 195) but in the proof the steps are: $T \geq C/(\lambda L) \geq 1/(\lambda L)^2$, which seems to suggest the condition is $T \geq C/(\lambda L)$ - is there a typo ? Furthermore, it is not clear why if the assumption does not hold, the lower-bound becomes vacuous ? Is it that the lower-bound of Theorem 8 then dominates the lower-bound of Theorem 9 ? In addition, the key step in line 207 seems to be that $|L-\hat{L}|/L| \leq 1$, which is not the step that is explained and seems non-trivial. In line 211 a square root appears on the ln term in the second inequality but is not present in the inequality before and after ? And then the final result in Theorem 9 has a square root on the ln term again making things quite confusion (and this seems to contribute to the confusion around the bound becoming vacuous compared to Theorem 8). Finally, beside the issue around the square-root on the ln term, it seems that Theorem 9 follows from line 211 by taking the average of the final two inequalities (of line 211) and if so this could be explicitly stated for clarity. Overall, this part is quite confusing and could be made clearer.
- In line 224-225, you use the notation $x_t$ to discuss the definition of $B_\gamma^K$ but $x_t$ is the notation used for the point played by the algorithm at time step $t$. It is unclear why this notation is used here (similarly for $\nabla f^{\circ}(x_t)$ in the statements of Theorem 8 and 9).
- In line 249, the assumption on the loss vectors refers to the growth condition assumptions from line 42 ? If so, this could be made clearer, for example just adding “growth” in front of assumptions. This is discussed 2 paragraphs down but it is not necessarily clear at this point.
- Line 251-254: am I correct in understanding that your approach guarantees a $O(\sqrt{T})$ regret even in the adversarial setting because it satisfies the universal online learning condition ? If so, it would be nice to make this more explicit.

Typos:
- Line 222 missing a “the” in front of inequality.
- Line 219, missing a “the” in front of ball.
- Line 245: typo “$x_\star$ in on corners" ?
- Line 30, the definition of the regret $R_T$ is written with $f_t(x)$ and not $f_t(x_\star)$. A similar point for line 102 - perhaps defining the regret directly w.r.t. $x_\star$ would make the notation clearer instead of with respect to an arbitrary point $x$ without making this dependence explicit in the notation $R_T$.
- line 147: missing "a" in front of "convex body K".
- Line 287: missing “the” in front of ball.
- Line 286: "for this $\tilde{B}^K_\gamma$" is at the start of a sentence but not capitalised and $\tilde{B}^K_\gamma$ is not defined yet.
- Line 20: you state that uniformly convex sets include $\ell_p$-balls for $p \in [1, \infty)$ but this should be for $p \in [2, \infty)$ I think. (same issue in line 78).

**Questions:**

Some questions are asked in the weaknesses above.
- In Corollary 12, do you think the dependence on $GD$ can be removed? Would it be possible to adjust the lower-bound so that these also appear ?
- The upper bound in Theorem 10 is shown to be tight in the case of $K = W_\lambda$ with $d=2$. Is there any dimension dependence that is hidden in some of the terms in the bound of Theorem 10 ? And could these be shown to be necessary by considering the lower-bound on $W_\lambda$ in higher dimensions ?
- For the result on uniformly convex sets, it is mentioned that a similar analysis is possible for corrupted stochastic environments. Without necessarily giving the details of the analysis (which I realise might be quite similar to the one already given), it would be nice to include a statement of the result in this case or at least what the dependence on the corruption level $C$ is in the bound and a bit of a discussion of this. In particular so that the reader can get an idea of the result in the fully adversarial setting when $C = T$ (which is the setting [10] consider it seems) and whether the $\log T$-$\sqrt{T}$ interpolation still holds in this case.

**Limitations:**

In general yes, though it is not necessarily apparent that in the fully adversarial setting, the results do not improve upon FTL.

---

> ### Author Rebuttal · Authors · 2024-08-07
>
> We appreciate your valuable time and thorough review. We will correct the minor comments and typos in the revised version. Here are our responses to the review.
>
> **Rebuttal to Weaknesses**
>
> > The improvement over FTL is mostly in the stochastic environment. In Theorem 13, if we consider the fully adversarial environment ($C=T$) for example, then the regret is $\sqrt{T}$ and the $\log T$ results for FTL are not improved on. Do you believe this is a fault in the analysis or that something beyond the universal online learning conditions are required ?
>
> When the environment is completely adversarial, it is unclear whether the regret of $\sqrt{T}$ is due to a fault in the analysis or if additional assumptions are needed for universal online learning.
> Exploring this is an important direction for future work.
> However, it is important to note that the conditions under which FTL achieves fast rates and the conditions under which universal online learning achieves fast rates are neither sufficient nor necessary for each other.
> FTL can achieve fast rates under the assumption of the growth condition $\\|g_1 + \dots + g_t\\| \geq tL$ for some $L > 0$, which is neither a sufficient nor necessary condition $\nabla f^\circ(x^*) \neq 0$, under which universal online learning to achieve fast rates.
>
> > Many of the results work with the $\ell_2$-norm and it is not clear what the results would be when working with respect to an arbitrary norm $\\|\cdot\\|$.
> In particular, it would be nice to at least have a discussion of if/how these results could be extended to arbitrary norms.
> It also seems that considering the Euclidean ball in Definition 5 means that $\ell_p$-balls for $p > 2$ are not sphere-enclosed sets. It would be good to have a discussion of the relationship between uniformly convex sets and sphere-enclosed sets as well as a discussion on how the results from Section 4.1 link to the results in Section 4.4.
>
> Extending the sphere-enclosed condition to an arbitrary norm $\\|\cdot\\|$ is challenging because the sphere-enclosed condition leverages the fact that the enclosing ball is an Euclidean ball.
>
> However, our result for uniformly convex sets (Theorem 15) holds for general norms, making this a very general result.
> It is worth noting that the result of Theorem 15 corresponds to the result for sphere-enclosed sets when $q = 2$ and $\\|\cdot\\|$ is the Euclidean norm.
> Additionally, Theorem 15 does not require the feasible set $K$ to be globally "curved."
> In fact, Theorem 15 only uses the inequality
> $\langle \nabla f^\circ(x\_\star), x - x\_\star \rangle \geq \frac{\kappa}{4} \\| x - x_\star \\|^q \cdot \\|\nabla f^\circ(x\_\star) \\|\_\star$ for all $x \in K$, which describes the local curvature around the optimal solution $x_\star$.
> In the revised version, we will discuss these points in detail.
>
> > The generality of the assumption of sphere-enclosure of the feasible set is well motivated (what is the mild condition on $\nabla f^\circ(x_\star)$ in line 231 ?). However, the assumption that $x_\star$ is on the boundary of $K$ could be discussed further.
> In particular, it is only justified for linear losses. Perhaps a slightly more developed discussion of this could help (or/and more pointers to the literature). Moreover, it would be nice to have an explanation of why this assumption is needed - is it just to guarantee?
>
> The mild condition on $\nabla f^\circ(x_\star)$ in line 231 refers to Condition (iii) of the sphere-enclosed set definition (Definition 5).
> When the feasible set $K$ is a polytope, if $\nabla f^\circ(x_\star)$ is aligned with the direction parallel to the face of the polytope passing through $x_\star$, it becomes impossible to enclose $K$ with a sphere passing through $x_\star$.
> Therefore, such a polytope is not sphere-enclosed.
> In the revised version, we will add an explanation with a figure to illustrate this point and clarify the description in line 231.
>
> Additionally, the assumption $\nabla f^\circ(x_\star) \neq 0$ is common in offline optimization, as discussed in lines 255-259. Although we do not yet have a complete intuition for this, we believe that $\\| \nabla f^\circ(x\_\star)\\|$ serves a similar role to the minimum suboptimality gap in the bandit problem.
> We will try to include it as much as possible in the revised version.
>
> > In the statement of Theorem 9, it is stated that $T \geq C / (\lambda L)^2$ (which is also stated in line 195) ..., is there a typo ?
>
> Thank you for pointing this out. We apologize for the typo; as the reviewer pointed out, it should be $T \geq C / (\lambda L)$.
>
> >  the key step in line 207 seems to be that $|L - \hat{L}| / L \leq 1$, which is not the step that is explained and seems non-trivial.
>
> From the discussion in lines 199-120, we have $L \geq \hat{L} (> 0)$. Therefore, $|L - \hat{L}| / L = L / L = 1$.
> In the revised version, we will clarify this point.
>
> > In line 211 a square root appears on the ln term in the second inequality but is not present in the inequality before and after ?
>
> Thank you for pointing this out. We apologize for the error; the square root in $\ln(\cdot)$ in the second inequality is unnecessary. We will correct this in the revised version.
>
> Due to space constraints, we will respond to the points that are considered to have a relatively small impact on the evaluation in the following Comments.

---

> > ### Comment · Reviewer_BHA1 · 2024-08-08
> >
> > Thanks to the authors for their response, which clarify some of the points I had raised. I am happy to keep my score.

---

> ### Author Response · Authors · 2024-08-07
> **Additional Replies**
>
> Here, we will respond to the points that are considered to have a relatively small impact on the evaluation, which cannot be included in the above rebuttal due to space constraints.
>
> > it seems that Theorem 9 follows from line 211 by taking the average of the final two inequalities (of line 211) and if so this could be explicitly stated for clarity
>
> Thank you for pointing this out. In the revised version, we will adjust the statement of the Theorem accordingly.
>
> > Furthermore, it is not clear why if the assumption does not hold, the lower-bound becomes vacuous ?
>
> We apologize for the inappropriate use of the term "vacuous". We will simply change the expression to "We assume that $T \geq C / (\lambda L)$".
>
> > In line 224-225, you use the notation $x_t$ to discuss the definition of $B_{\gamma}^K$ but $x_t$ is the notation used for the point played by the algorithm at time step. It is unclear why this notation is used here (similarly for $\nabla f^{\circ}(x_t)$ in the statements of Theorem 8 and 9).
>
> The use of $x_t$ here is because it will be used in Theorem 10, as stated in line 222, "We will see in the following proof that...".
> However, as you pointed out, the description in Remark 1 can be improved, and we will improve this in the revised version.
> Specifically, we will change it to " $x \in B_\gamma^K$ is equivalent to $\langle \nabla f^\circ(x_\star), x - x^* \rangle \geq \gamma \\| x - x^* \\|_2^2$".
> We apologize for the typo in Theorems 8 and 9 where $\nabla f^{\circ}(x_t)$ should be $\nabla f^{\circ}(x^*)$. Thank you for pointing this out.
>
> > - In line 249, the assumption on the loss vectors refers to the growth condition assumptions from line 42 ? If so, this could be made clearer, for example just adding “growth” in front of assumptions. This is discussed 2 paragraphs down but it is not necessarily clear at this point.
> >- Line 251-254: am I correct in understanding that your approach guarantees a
>  regret even in the adversarial setting because it satisfies the universal online learning condition ? If so, it would be nice to make this more explicit.
>
> Thank you very much for your valuable comments. We will prepare the revised version according to your comments.
>
> **Replies to Questions**
>
> > In Corollary 12, do you think the dependence on $GD$ can be removed? Would it be possible to adjust the lower-bound so that these also appear ?
> > The upper bound in Theorem 10 is shown to be tight in the case of $K = W_{\lambda}$ with $d = 2$. Is there any dimension dependence that is hidden in some of the terms in the bound of Theorem 10 ? And could these be shown to be necessary by considering the lower-bound on $W_\lambda$ in higher dimensions ?
>
> We are uncertain whether $GD$ can be eliminated from the regret upper bound or introduced into the regret lower bound.
> Regarding the dependency on $d$ in Theorem 10, we believe that it does not depend on $d$ if the Maler algorithm is used.
> This is because the constants $c_{\mathsf{sc}}$ and $c'_{\mathsf{sc}}$ in the Maler algorithm do not depend on $d$.
>
> > For the result on uniformly convex sets, it is mentioned that a similar analysis is possible for corrupted stochastic environments. Without necessarily giving the details of the analysis (which I realise might be quite similar to the one already given), it would be nice to include a statement of the result in this case or at least what the dependence on the corruption level $C$ is in the bound and a bit of a discussion of this.
>
> Thank you for your advice. In the revised version, we will also include results for the corrupted setting.

---

### Official Review · Reviewer_Mo5s · 2024-07-05

**Soundness:** 3
**Presentation:** 2
**Contribution:** 2
**Rating:** 5
**Confidence:** 4

**Summary:**

In this work, the authors study stochastic online convex optimization (SOCO), and establish a tighter regret bound of $O(\log T)$ for sphere-enclosed sets. Specifically, such a regret bound is derived by exploiting an immediate result of several existing universal online learning algorithms. Moreover, the authors also demonstrate that the immediate result can be utilized to recover the existing regret bound for SOCO over strongly convex sets, and could improve the existing regret bound over uniformly convex sets. In addition, a corrupted setting is also considered, and lower bounds for non-corrupted and corrupted are also provided.

**Strengths:**

This paper studies stochastic online convex optimization (SOCO), and mainly focuses on feasible sets with special curvature. Although this paper is not the first to study the problem, it has the following strengths compared with previous studies.
1) Different from previous studies that need the strong convexity of sets to achieve an $O(\log T)$ regret bound for SOCO, this paper introduces a new condition on the set to obtain the $O(\log T)$ regret bound, i.e.,  sphere-enclosed sets.
2) For SOCO over the uniformly convex sets, this paper has established a better regret bound than existing results.
3) The authors also consider a corrupted setting that interpolates the fully adversarial and stochastic settings.
4) Matching Lower bounds for a special case of ellipsoid sets are provided for the non-corrupted and corrupted settings.

**Weaknesses:**

Although this paper provides some new results for stochastic online convex optimization (SOCO), I have some concerns. First, compared with stochastic (offline) optimization, the main challenge of online convex optimization (OCO) is how to deal with a fully adaptive adversary. However, this paper mainly focuses on SOCO, where the adversary is even weaker than the oblivious one, which is not very interesting to me.

Second, although it seems that this paper has made many contributions, some results have already been achieved, and some results can be simply derived.
1) The authors argue that previous studies [8,16] only consider online linear optimization (OLO) that may prevent us from leveraging the curvature of loss functions, and the existing FTL algorithm [8] will suffer an $\Omega(T)$ regret if the ideal condition is not satisfied. However, in Section 4 of the existing work [8], Huang et al. have already proposed an adaptive algorithm to avoid the $\Omega(T)$ regret of FTL, whose regret is at most $O(\sqrt{T\log T})$. The main idea is to utilize an expert tracking algorithm to combine FTL with an optimal algorithm for the general OCO. I also refer the authors to a missing related work [*], which provides an online lazy gradient descent algorithm, which enjoys the same regret bound as the algorithm of Huang et al.
2) Note that universal online learning algorithms utilized in this work follow a very similar idea of using the expert tracking algorithm. Moreover, it is easy to combine FTL with universal online learning algorithms. Without the analysis provided in this paper, one can simply prove that such a combination can achieve the $O(log T)$ regret of FTL for SOCO under ideal conditions, and leverage the curvature of loss functions (and the convexity) to recover the optimal regret bounds for general OCO.
3) The derivation of the lower bound in Theorem 8 seems to be very simple, i.e., just computing and replacing a constant of an existing lower bound with the gradient norm in the fixed optimal solution. Moreover, the extensions (about both the lower and upper bounds) to the corrupted setting do not bring noteworthy challenges.
4) The sphere-enclosed sets seem to be a bit complicated. Moreover, as discussed by the authors it is very similar to the Bernstein condition, which is more common and has been used to achieve a tighter regret bound for SOCO. So, it is not surprising that the sphere-enclosed sets can also be used to derive similar results.

Third, the writing and organization of this paper are not very sound.
1) The context of Section 2.1 is almost the same as the first paragraph in Section 1.
2) Many short proofs are interluded in the main text, which reduces the readability of this work.
3) In line 340, "a similar analysis ideas" seems to be a typo.

[*] Anderson and Leith. Online Lazy Gradient Descent is Universal on Strongly Convex Domains. In NeurIPS 2021.

----After Rebuttal----
After reading the authors' response, I realize that it is very hard (and even impossible) to combine FTL with an existing universal online learning algorithm. So, the analysis of universal online learning algorithms provided in this paper is valuable, and I increase my score to 5.

**Questions:**

Besides the weaknesses discussed above, I also have the following questions.
1) Can the authors summarize the fully new contributions compared with existing studies?
2) In Theorem 14, the authors exploit the curvature of loss functions and feasible sets simultaneously. Are those regret bounds better than the existing regret bounds derived by only exploiting the curvature of loss functions?

**Limitations:**

The authors have discussed some limitations of this work.

---

> ### Author Rebuttal · Authors · 2024-08-07
>
> Thank you very much for your valuable time to give the insightful review. Below are our responses to the review.
>
> **Rebuttal to Weaknesses**
>
> > First, compared with stochastic (offline) optimization, the main challenge of online convex optimization (OCO) is how to deal with a fully adaptive adversary. However, this paper mainly focuses on SOCO, where the adversary is even weaker than the oblivious one, which is not very interesting to me.
>
> We acknowledge that our approach cannot achieve a fast rate in fully adversarial environments.
> Still, note that our approach can achieve fast rates when the environment is stochastic while achieving $O(\sqrt{T})$ in adversarial environments.
> Furthermore, please note that the conditions under which Follow the Leader (FTL) achieves fast rates are neither sufficient nor necessary for the conditions under which universal online learning achieves fast rates.
> Investigating whether it is possible to achieve fast rates similar to FTL using universal online learning in adversarial environments is an important direction for future work.
>
> > ... in Section 4 of the existing work [8], Huang et al. have already proposed an adaptive algorithm to avoid the $\Omega(T)$ regret of FTL, whose regret is at most $O(\sqrt{T \log T})$. The main idea is to utilize an expert tracking algorithm to combine FTL with an optimal algorithm for the general OCO.
> I also refer the authors to a missing related work [*], ..., which enjoys the same regret bound as the algorithm of Huang et al.
> > Anderson and Leith. Online Lazy Gradient Descent is Universal on Strongly Convex Domains. In NeurIPS 2021.
>
>
> We apologize for not noticing and not mentioning Section 4 of Huang et al. [8] in the main text.
> In the revised version, we will appropriately address this and revise the explanation in the introduction.
>
> We also apologize for overlooking the paper by Anderson and Leith (2021). Thank you for bringing it to our attention.
> In the revised version, we will thoroughly review its content and cite it appropriately.
>
> > Note that universal online learning algorithms utilized in this work follow a very similar idea of using the expert tracking algorithm.
>
> Thank you for your insightful feedback.
> We will mention the expert tracking algorithm in the revised version.
> Still, please note that our research focuses on identifying new conditions under which universal online learning can be accelerated, and we are not proposing a new algorithm.
>
> > it is easy to combine FTL with universal online learning algorithms. Without the analysis provided in this paper, one can simply prove that such a combination can achieve the regret of FTL for SOCO under ideal conditions, ...
>
> Thank you for your valuable comments.
> In fact, we were unable to combine FTL with universal online learning. If you have time, could you please explain how they can be combined?
> Additionally, if there are any relevant references, we would appreciate it if you could share them.
> In the framework of universal online learning (e.g., MetaGrad), different learning rates are set for the base learners and then integrated. Should we introduce a base learner corresponding to $\eta \to \infty$ for FTL?
>
> Still, please note that the conditions under FTL achieves fast rates, $\|g_1 + \cdots + g_t \| \geq t L$ for some $L > 0$, are neither sufficient nor necessary for the conditions under which universal online learning achieves fast rates, $\nabla f^\circ(x_\star) \neq 0$.
> Therefore, even if FTL could be incorporated into universal online learning, it does not diminish the value of our contribution.
>
> > as discussed by the authors it is very similar to the Bernstein condition, which is more common and has been used to achieve a tighter regret bound for SOCO.
>
> As you pointed out, when loss functions are linear, online linear optimization over a sphere-enclosed set satisfies the Bernstein condition, as discussed in Appendix H.
> Hence, our research can also be seen as identifying new conditions under which the Bernstein condition is satisfied.
> However, it is important to note that the sphere-enclosed condition can only be reduced to the Bernstein condition when the underlying loss functions are linear, while our results apply to convex functions that are not necessarily linear.
>
> **Replies to Questions**
> > Can the authors summarize the fully new contributions compared with existing studies?
>
> Our contribution lies in identifying new conditions under which the framework of universal online learning can achieve fast rates.
> Specifically, we introduced the new characterization of feasible sets called sphere-enclosed sets and demonstrated that fast rates can be achieved on these sets by universal online learning.
> Online linear optimization over the sphere-enclosed set can be reduced to the Bernstein condition in the context of online linear optimization.
> Still, our results hold for more general convex loss functions.
> Furthermore, by extending the analysis for sphere-enclosed sets, we showed that universal online learning can achieve fast rates for more general sets, such as strongly convex sets and uniformly convex sets, and this result matches the existing lower bound.
>
> > In Theorem 14, the authors exploit the curvature of loss functions and feasible sets simultaneously. Are those regret bounds better than the existing regret bounds derived by only exploiting the curvature of loss functions?
>
> When the curvature of the feasible set is greater than the curvature of the loss functions, our results improve upon the existing regret upper bounds that only utilize the curvature of the loss functions.
>
> Due to space constraints, we will respond to the points that are considered to have a relatively small impact on the evaluation in the following Comments.

---

> ### Author Response · Authors · 2024-08-07
> **Additional Replies**
>
> Here, we will respond to the points that are considered to have a relatively small impact on the evaluation, which cannot be included in the above rebuttal due to space constraints.
>
> > Third, the writing and organization of this paper are not very sound.
> > 1. The context of Section 2.1 is almost the same as the first paragraph in Section 1.
> > 2. Many short proofs are interluded in the main text, which reduces the readability of this work.
> > 3. In line 340, "a similar analysis ideas" seems to be a typo.
>
> Regarding point 1, I believe it is quite common to restate the problem formulation in the preliminaries section following the introduction.
>
> For point 2, if you have time, could you please specify which parts you found to have low readability if the reviewer has time? We will make an effort to reduce the number of inline equations in the revised version.
>
> Regarding point 3, thank you for pointing this out. We will correct it in the revised version.

---

> ### Comment · Reviewer_Mo5s · 2024-08-08
>
> Thanks for the authors' response. I currently realize that it is very hard (and even impossible) to combine FTL with an existing universal online learning algorithm. So, the analysis of universal online learning algorithms provided in this paper is valuable, and I will increase my score from 3 to 5.

---

### Official Review · Reviewer_kmSZ · 2024-07-13

**Soundness:** 3
**Presentation:** 3
**Contribution:** 3
**Rating:** 6
**Confidence:** 3

**Summary:**

The paper relates the regrets of (stochastic) online convex optimization to the geometry of the feasible set. Precisely, the paper proposes the sphere-enclosed-feasible-set property which measures the local curvature condition of the feasible set at the offline optimal solution, and shows that many existing online convex optimization algorithms, under this property, achieves regret O(\rho*\log(T)), the fast rate. In addition, similar regret upper bounds are shown to be valid in various settings, including the stochastic with corruption, convex (nonlinear) objectives, etc.

**Strengths:**

The introduced sphere-enclosed-feasible-set property appears innovative, and provide insights into conditions under which fast rate is achievable.

**Weaknesses:**

The paper shows that many existing algorithms can converge at the fast rate under the proposed sphere-enclosed-feasible-set condition. Thus, one might argue that there is not too much improvement over previous works on the algorithmic side. In addition, the lower and upper bounds provided appear to hold under slightly different conditions (see the question below).

**Questions:**

I’m wondering if there is any connection between the \lambda-strongly convex feasible set and the sphere-enclosed feasible set? The lower bounds are provided for the case when the feasible sets are ellipsoid or \lambda-strongly convex, but the upper bounds are provided under the sphere-enclosed condition. It would be great to compare the lower and upper bound under the same set of conditions.

**Limitations:**

Yes, the paper points out that one limitation is that the main results hold under stochastic environment.

---

> ### Author Rebuttal · Authors · 2024-08-07
>
> We are grateful for your valuable time and insightful review.
> Below are our responses to the review.
>
> > In addition, the lower and upper bounds provided appear to hold under slightly different conditions (see the question below).
>
> and
>
> > I’m wondering if there is any connection between the \lambda-strongly convex feasible set and the sphere-enclosed feasible set? The lower bounds are provided for the case when the feasible sets are ellipsoid or \lambda-strongly convex, but the upper bounds are provided under the sphere-enclosed condition. It would be great to compare the lower and upper bound under the same set of conditions.
>
> Thank you for pointing out this important point.
> The upper bound for sphere-enclosed sets (Theorem 10) corresponds to the lower bound for the ellipsoid $W_{\lambda}$ (Theorem 8). This is because, as stated in Proposition 11, the ellipsoid $W_{\lambda}$ is sphere-enclosed with $\rho = 1 / \lambda$.
>
> The upper bound for uniformly convex sets (Theorem 15) corresponds to the lower bound for the ellipsoid $W_{\lambda}$ (Theorem 8). This is because uniformly convex sets include strongly convex sets as a special case, and $W_{\lambda}$ is $\lambda$-strongly convex with respect to $\\|\cdot\\|_2$.
>
> However, as the reviewer pointed out, our paper does not provide a lower bound for general sphere-enclosed feasible sets.
> In the revised version, we will emphasize this point and also note that the ellipsoid $W_{\lambda}$ is sphere-enclosed with $\rho = 1 / \lambda$.

---

> > ### Comment · Reviewer_kmSZ · 2024-08-08
> >
> > Thank the authors for additional comments on lower and upper bounds, which answer my questions. I would like to keep my score!

---

### Official Review · Reviewer_b39D · 2024-07-13

**Soundness:** 3
**Presentation:** 3
**Contribution:** 3
**Rating:** 7
**Confidence:** 3

**Summary:**

This paper considers the influence of feasible set geometry on the OCO regret. For a class of algorithms satisfying certain regret upper bounds (including MetaGrad and Maler), the authors provide new fast-rate results in the stochastic convex loss setting and the corrupted convex loss setting. Their results also match existing lower bound results.

**Strengths:**

Most contributions of this paper are new in the OCO area, including:

1.A novel generalization of previous curvature-dependent regret for linear optimization to general stochastic convex loss.

2.New regret results in the corrupted setting.

3.A new result that simultaneously utilizes the curvature of the loss and the feasible set.

**Weaknesses:**

One possible limitation of the results is that, unlike the OLO results, the action set curvature-dependent rates for the convex losses in this paper don't hold for the adversarial setting.

**Questions:**

What's the main technical challenge in extending existing results for OCO from the stochastic setting to the stochastically extended adversary (SEA) setting or even the adversarial OCO setting? Could the authors explain more about the gap in simply applying the linearization technique in this setting?

**Limitations:**

N.A.

---

> ### Author Rebuttal · Authors · 2024-08-07
>
> Thank you for your valuable time and careful review. We will address the minor comments and typos in the revised version. Below are our responses to the review.
>
> **Rebuttal to Weaknesses**
>
> > One possible limitation of the results is that, unlike the OLO results, the action set curvature-dependent rates for the convex losses in this paper don't hold for the adversarial setting.
>
> We acknowledge that this is indeed a shortcoming of our approach, as you pointed out.
> Still, note that our approach can achieve fast rates when the environment is stochastic while achieving $O(\sqrt{T})$ in adversarial environments.
> Furthermore, please note that the conditions under which Follow the Leader (FTL) achieves fast rates are neither sufficient nor necessary for the conditions under which universal online learning achieves fast rates.
> Investigating whether it is possible to achieve fast rates similar to FTL using universal online learning in adversarial environments is an important direction for future work.
>
> **Replies to Questions**
>
> > What's the main technical challenge in extending existing results for OCO from the stochastic setting to the stochastically extended adversary (SEA) setting or even the adversarial OCO setting? Could the authors explain more about the gap in simply applying the linearization technique in this setting?
>
> Thank you for the question.
> At this point, we are uncertain whether an extension to the SEA model is possible, and exploring this is an important direction for future work.

---

> > ### Comment · Reviewer_b39D · 2024-08-12
> >
> > Thank you for your response. I will keep my score positive.

---

### Decision · Program_Chairs · 2024-09-25

**Decision:**

Accept (poster)

**Comment:**

This paper considers how to achieve fast rates in stochastic environments of online convex optimization by exploiting the curvature of feasible sets. The authors present various interesting results, including leveraging the nice results from universal online learning. All the reviewers unanimously appreciate these new contributions!

Nevertheless, one (expert) reviewer raised valid concerns about the paper presentation and also proposed several technical discussions, such as the potential triviality of combining FTL with existing universal online algorithms (though we all realized that it is not easy to achieve so after back-and-forth discussions). The authors are encouraged to revise the manuscript to reflect this point and incorporate all the constructive comments from the reviewers.

Two additional comments after my own reading of the paper:

- The NeurIPS’16 paper "Combining Adversarial Guarantees and Stochastic Fast Rates in Online Learning" also provides fast-rate implications for stochastic OCO/PEA with the "second-order regret bound" (achieved by MetaGrad or Squint), which should be discussed more carefully.

- It's interesting to see the authors considering extending these results to the SEA model. The gradient-variation universal online learning algorithms (such as NeurIPS’23 Universal Online Learning with Gradient Variations) could be useful.